# Privatisation rescues function following loss of cooperation

Sandra Breum Andersen[1,2†]*, Melanie Ghoul[1], Rasmus L Marvig[3], Zhuo-Bin Lee[1], Søren Molin[2], Helle Krogh Johansen[4,5], Ashleigh S Griffin[1]*

[1]Department of Zoology, University of Oxford, Oxford, United Kingdom; [2]Novo Nordisk Foundation Center for Biosustainability, Technical University of Denmark, Lyngby, Denmark; [3]Center for Genomic Medicine, Rigshospitalet, Copenhagen, Denmark; [4]Department of Clinical Microbiology, Rigshospitalet, Copenhagen, Denmark; [5]Department of Clinical Medicine, Faculty of Health and Medical Sciences, University of Copenhagen, Copenhagen, Denmark

**Abstract** A single cheating mutant can lead to the invasion and eventual eradication of cooperation from a population. Consequently, cheat invasion is often considered equal to extinction in empirical and theoretical studies of cooperator-cheat dynamics. But does cheat invasion necessarily equate extinction in nature? By following the social dynamics of iron metabolism in *Pseudomonas aeruginosa* during cystic fibrosis lung infection, we observed that individuals evolved to replace cooperation with a 'private' behaviour. Phenotypic assays showed that cooperative iron acquisition frequently was upregulated early in infection, which, however, increased the risk of cheat invasion. With whole-genome sequencing we showed that if, and only if, cooperative iron acquisition is lost from the population, a private system was upregulated. The benefit of upregulation depended on iron availability. These findings highlight the importance of social dynamics of natural populations and emphasizes the potential impact of past social interaction on the evolution of private traits.

DOI: https://doi.org/10.7554/eLife.38594.001

*For correspondence:
sandrabreumandersen@gmail.com (SBA);
ashleigh.griffin@zoo.ox.ac.uk (ASG)

Present address: [†]Langone Medical Center, New York University, New York, United States

Competing interests: The authors declare that no competing interests exist.

Reviewing editor: Otto Cordero,

## Introduction

Identifying mechanisms responsible for the maintenance of cooperation is a major achievement of evolutionary biology (*Axelrod and Hamilton, 1981*; *Hamilton, 1963*). We can predict in which conditions cooperation will thrive, and where it might pay to exploit cooperative neighbours. Evidence of tension between cooperative and cheating strategies are all around us in nature – for example, in the co-evolution between flowers and their pollinators (*Jandér and Herre, 2010*), the policing and counter-policing behaviours of social insects (*Foster and Ratnieks, 2000*) – and in human society (*Mathew and Boyd, 2011*; *Rand et al., 2011*). Cases where these dynamics have resulted in the loss of cooperation are much less well understood, primarily because long-term consequences of cheat invasions are unobserved and unreported (*Sachs and Simms, 2006*). This is for the obvious reason that it is inherently difficult to detect a history of something that no longer exists. And also because it is generally the case that once cheating has gone to fixation, it is almost impossible for cooperation to re-invade (*Axelrod and Hamilton, 1981*). This is where most studies of social dynamics end.

Individuals are nevertheless under selection to survive the loss of cooperation. Consider a population where cheating has gone to fixation. If cooperation fulfilled an important function, this population may now be at risk of extinction (*Fiegna and Velicer, 2003*). To escape this fate, selection may favour individuals that can restore function by employing a 'privatisation' strategy – replacement of a mechanism that was once performed cooperatively as a group, with a selfish one that only benefits the actor (*Bel, 2006*). As such, we use the term privatisation as it is normally understood in common

language to describe a switch in strategy from one that helps a whole group to achieve a goal to one where a goal is achieved by the actor alone. This differs from its use for describing the acquisition of property for future benefit by a group or an individual (*Strassmann and Queller, 2014*), the retention of public goods for individual use (*Asfahl and Schuster, 2017*), and for when public goods only are useable to a part of a community (*Niehus et al., 2017*). Privatisation could be a common occurrence in nature that has nevertheless been overlooked because there is no reason, *a priori*, to interpret private function as a result of past social interaction. It may also have been missed for the practical reason that we are required to track behaviour over many generations post-cheat invasion in a natural population. We overcome these difficulties by studying the evolutionary dynamics of a cooperative trait for more than an estimated 1.3 million generations (*Supplementary file 1A*), in a population of bacterial cells. We report the first observation (to our knowledge) of a natural population responding to cheat invasion by adopting a privatisation strategy and, therefore, avoiding the possibility of extinction.

Our study population is comprised of *Pseudomonas aeruginosa* bacteria causing lung infection in patients with cystic fibrosis (CF). CF is a genetic disease that causes the build-up of dehydrated mucus in lungs and sinuses, which *P. aeruginosa* infects (*Folkesson et al., 2012*). The host actively withholds iron to limit infection, and, as a counter-measure, *P. aeruginosa* employ a range of different iron acquisition strategies (*Poole and McKay, 2003*). The primary mechanism relies on secretion of the siderophore pyoverdine (*Haas et al., 1991*; *Konings et al., 2013*). This has been demonstrated to be a cooperative trait, where iron-bound pyoverdine molecules are available for uptake not only by the producer, but also by neighbouring cells (*Buckling et al., 2007*; *West and Buckling, 2003*). *P. aeruginosa* also make a secondary siderophore, pyochelin, which is cheaper to produce but has a lower affinity for iron (*Dumas et al., 2013*). In contrast, the Pseudomonas heme uptake system (*phu*) is private, as the iron-rich compound heme is taken up directly without the secretion of exploitable exoproducts (*Table 1*). Additional mechanisms of uptake include the private ferrous iron transport system (*feo*; *Table 1* [*Cartron et al., 2006*]), and the heme assimilation system (*has*). The *has* system produces a hemophore that can bind to heme, or hemoglobin that contains heme. In the closely related system of *Serratia marcescens* it is conditionally cooperative, as uptake is direct without the hemophore at high concentrations (*Ghigo et al., 1997*; *Létoffé et al., 2004*).

*P. aeruginosa* iron metabolism evolves during long term infection. Pyoverdine production has repeatedly been found to be lost during CF infection (*Andersen et al., 2015*; *Jeukens et al., 2014*; *Konings et al., 2013*; *Martin et al., 2011*), and the *phu* system has been observed to be upregulated late in infection (*Marvig et al., 2014*; *Nguyen et al., 2014*). A major challenge is to identify the selective pressures experienced in situ that cause such changes. Host adaptation, to accommodate for example antibiotic pressure and resource limitations, is frequently predicted to be the

**Table 1.** Main iron uptake systems of *P. aeruginosa*.

The pyoverdine and pyochelin systems produce siderophores which can steal $Fe^{3+}$ from host carriers. The conditionally cooperative *has* system produces a hemophore that can bind to heme, or hemoglobin that contains heme. The *phu* and *feo* systems are private. All systems have a specific receptor and uptake feeds back to increase system expression. The illustration shows a receptor in the cell membrane taking up an iron source (star). In a cooperative system a secreted public good (circle) binds the target and is taken up by the receptor, whereas a private system takes up the iron source directly.

| System | Mechanism | Social status | Target | Illustration |
|---|---|---|---|---|
| Pyoverdine | Public good; siderophore and specific receptor | Cooperative | $Fe^{3+}$, host-bound | |
| Pyochelin | Public good; siderophore and specific receptor | Cooperative | $Fe^{3+}$, host-bound |  |
| *has* | Public good; hemophore and specific receptor | Conditionally cooperative or private | Heme, hemoglobin | |
| *phu* | Specific receptor | Private | Heme, hemoglobin | |
| *feo* | Specific receptor | Private | $Fe^{2+}$ | |

DOI: https://doi.org/10.7554/eLife.38594.002

primary force in pathogen evolution (*Lieberman et al., 2011*; *Young et al., 2012*). Experimentally recreating host conditions in vitro is challenging, given the complex interplay between spatial structure, host immunity and nutrient availability (*Clevers, 2016*), which we are only beginning to be able to measure (*Koo et al., 2011*). Clinical measurements may differ from experimental in vivo and in vitro findings (*Cornforth et al., 2018*), and studies using 'simple' experimental conditions do not always give comparable results to those that use more complex, and potentially more clinically relevant conditions (*Harrison et al., 2017*). In contrast, longitudinal sampling of patients to track in situ evolution provides an opportunity to infer selection without making *a priori* assumptions about the experienced environment. Evolutionary theory lets us make testable predictions to distinguish the effect of host adaptations and social interactions. With this approach we have shown that cheating is a major selective force in the loss of cooperative iron acquisition in the CF lung (*Andersen et al., 2015*).

Here, we identified changes in iron uptake strategies from genome sequences and correlated these with the social environment (the *P. aeruginosa* clonal lineage infecting a host), and duration of infection. Specifically, we test two hypotheses: first, we ask whether an increase in cooperation facilitates the breakdown of cooperation in a natural population, and second, can cells survive the loss of cooperation by switching to a private, and, therefore, unexploitable mechanism for acquiring iron. Two collections of *P. aeruginosa* samples were used, covering 551 whole genome sequenced isolates from 64 Danish patients, of 55 clone types (*Andersen et al., 2015*; *Markussen et al., 2014*; *Marvig et al., 2013*; *Marvig et al., 2015*; *Rau et al., 2012*; *Yang et al., 2011*); allowing for the identification of patterns of convergent evolution. While the *P. aeruginosa* populations within patients are remarkably uniform at the clone type level, colonizers diversify during infection (*Mowat et al., 2011*; *Winstanley et al., 2016*), and lineages may occupy distinct niches at late stages (*Jorth et al., 2015*). The isolate collection has been shown to represent the genetic diversity well, with multiple genotypes co-occurring within individual patient samples (*Sommer et al., 2016*). A key step in our analysis is to categorise isolates with respect to their social environment, for example, presence vs absence of cooperators (pyoverdine producers). This is necessary for testing our hypothesis that iron acquisition strategy is influenced by the social environment. The likelihood that we do not capture all co-infecting strains in our sample, for example leading to mis-classification of isolates as belonging to an environment without pyoverdine, will homogenise the sample groups and hence obscure any differences. Sampling errors of this kind will, therefore, hinder our ability to detect an effect (Type I error).

## Materials and methods

### Isolate collections

Two collections of whole genome sequenced *P. aeruginosa* isolates were used, as described previously (*Andersen et al., 2015*). The transmissible clonetypes DK1 (46 isolates) and DK2 (54 isolates) were sampled from 28 CF patients between 1972 and 2012 (*Jelsbak et al., 2007*; *Markussen et al., 2014*; *Marvig et al., 2013*; *Rau et al., 2012*; *Yang et al., 2011*). Six previously unanalysed DK1 isolates were included (from patients P55M3, P24M2 and P30M0, *Supplementary file 1B*). Further, 451 isolates of 54 clonetypes from 36 young CF patients were used (the 'children's collection' (*Andersen et al., 2015*; *Marvig et al., 2015*), of which 10 were of the DK1 clone type). At each sampling time, 1–8 colonies were cultured from patient sputum, and stored at −80° C (*Johansen et al., 2012*; *Sommer et al., 2016*). The sampling regimen for each patient is described in *Supplementary file 1B*, and *Appendix 1—figures 1* and *2*. The two collections complement each other as one covers a long time period, with relatively few samples from each patient and of both early and late infection stages, whilst the other covers the first 10 years of infection by environmental clone types, with more extensive sampling within patients. In the children's collection there is limited transmission between patients, but some clone types infect multiple patients, and some patients are infected by multiple clone types either at the same time or in succession (*Marvig et al., 2015*). All but two patients, however, harbour only one dominant clone type at a time and the social environment is thus made up of clonal lineages sharing the same ancestor. The analysis of increased pyoverdine production was only performed on the children's collection, as described below.

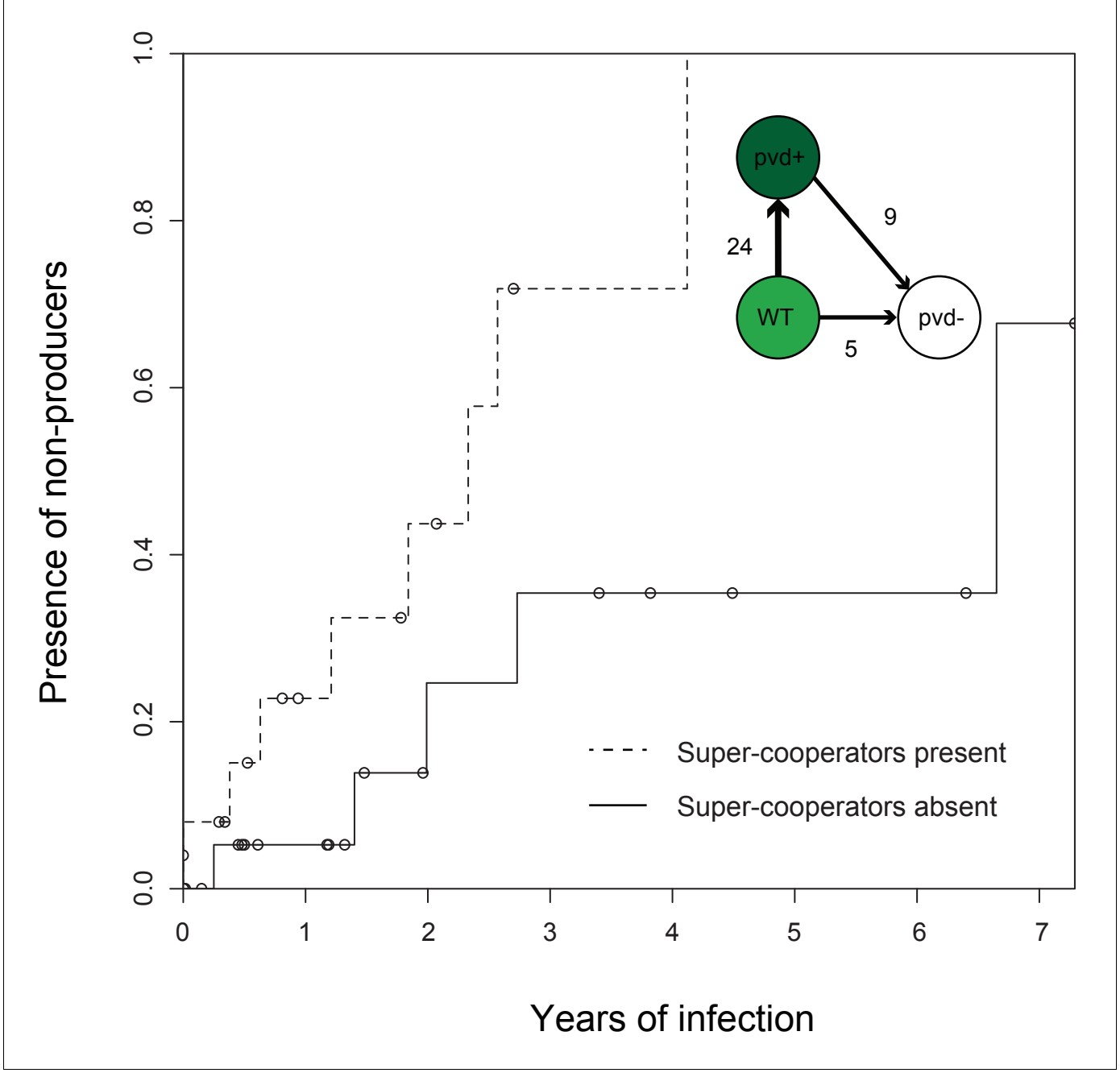

**Figure 1.** Super-cooperators precede the sampling of non-producers. Inverse Kaplan-Meier graph showing that pyoverdine non-producers are found more often in the presence of super-cooperators (dotted line, n = 25 representing 24 clonal lineages in patients, one of which had two independent losses of pyoverdine production), compared to when super-cooperators are absent (solid line, n = 22, representing 21 clonal lineages in patients, one of which had two independent losses). Circles indicate when sampling of clonal lineages stopped, without the observation of lost pyoverdine production. Insert shows that a transition from wild-type (WT) production to super-cooperation (pvd+) was observed in 24 clonal lineages, followed by nine independent occurrences of non-producers. Non-producers evolved in the absence of super-cooperators five times. In total, 33 clone types infecting 35 patients were followed.

DOI: https://doi.org/10.7554/eLife.38594.003

The phylogeny of the 54 DK2 isolates has previously been reported by *Marvig et al. (2013)*, and this was adapted for this study to phylogenetically map the order of mutations in regions of interest. To allow a similar analysis for the DK1 clone type, we used Snippy (*Seeman, 2015*) to identify SNPs in each of the 56 DK1 isolates relative to the PAO1 reference genome (RefSeq assembly accession

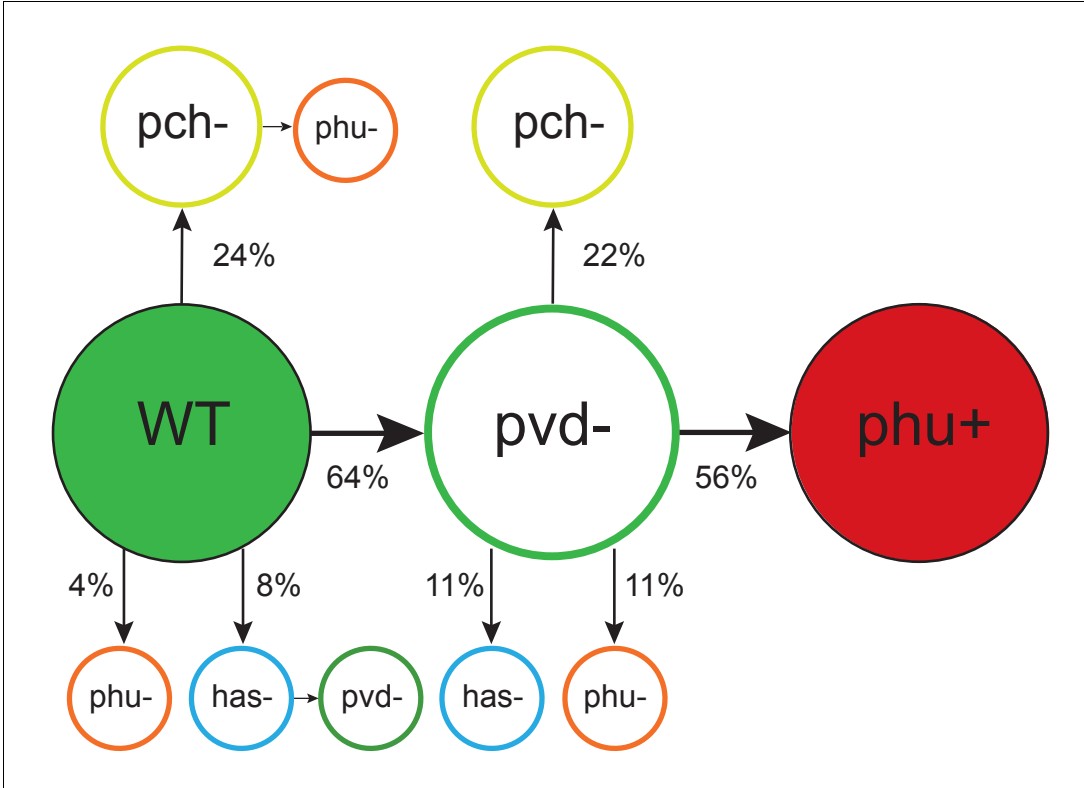

**Figure 2.** Order of mutations across iron acquisition systems. Wild type-like (WT) isolates colonize patients, and subsequent loss of cooperative pyoverdine production (pvd-) is the most common change in iron metabolism (n = 25), compared to mutations expected to affect the pyochelin, *phu* and *has* systems negatively (pch-, phu-, has-). Loss of pyoverdine production in clonal lineages is followed by intergenic *phuS*//*phuR* mutations, predicted to upregulate the private *phu* system, significantly more often than expected by chance (phu+, n = 9). Colour indicates iron uptake system (green = pyoverdine, yellow = pyochelin, dark red = intergenic *phuS*//*phuR*, orange = *phu* (either a mix of *phuS*//*phuR* and other *phu* mutations, or only ns SNPs), blue = *has*). The occurrence of *has* mutations may be underestimated due to low sequencing depth of two genes. The figure shows only transitions where there was a clear order in which systems were affected, see **Supplementary file 1F** for all transitions.

DOI: https://doi.org/10.7554/eLife.38594.004

GCF_000006765.1) and compared the SNPs of all the DK1 isolates to identify those that differed between the isolates. The SNPs were further filtered to only retain SNPs at positions covered by at least 10 reads in all isolates and exclude those where all isolates show more than 80% non-reference reads at the given position. Based on 4,639 SNPs within 2,614,892 sites, a maximum-likelihood phylogenetic tree was computed with RAxML version 8.2.11 (*Stamatakis, 2014*), with default settings using a general time reversible model of nucleotide substitution (option -m GTRCAT). The tree was rooted with the PAO1 reference genome.

The length of infection of each clone type in each patient was calculated. Clone types sampled only once from an individual patient were excluded (23 isolates of 22 clone types). For the transmissible clone types DK1 and DK2 the total length of infection across patients was used (39 and 36 years, respectively). This is a very conservative estimate of length of infection, as evolution occurs independently in different patients. To calculate the number of generations, a doubling time of 74 min was used for clonal lineages infecting for less than 6 years, 83 min for 6–24 years and 109 min for >24 years based on the measurements of *Yang et al. (2011)*.

## Changes in pyoverdine production during infection

We tested if the appearance of pyoverdine non-producers is correlated with preceding upregulation. Measurement of pyoverdine production was carried out by fluorescence readings at 400/460 nm

excitation/emission with a 475 nm cut-off following growth in iron-limited media (*Andersen et al., 2015*; *Kümmerli et al., 2009*). For each clonal lineage the baseline pyoverdine production was determined as that of the first isolate(s) of a clone type from a patient, and the production of all subsequent isolates of this clone type was compared to this. An isolate was classified as an over-producer if its production was >30% higher than that of the first. This cut-off was chosen as ~95% of the isolates had a lower standard deviation between replicates. The data is available from Dryad (doi:10.5061/dryad.6963pj3). The analysis was performed on longitudinally sampled clone types from the children's collection only, as the DK1 and DK2 isolates were not sampled frequently within patients. Only non-producers evolved from a clonal lineage that produced pyoverdine at the beginning of the infection period were included. Therefore clone types DK1 (10 isolates, P36F2), DK11 (two isolates, P51M5), DK40 (two isolates, P73M4) and DK53 (10 isolates, P36F2) were excluded from the analysis, as they were 'chronic' clone types with no pyoverdine producers sampled from the young patients (*Appendix 1—figure 1*; *Andersen et al., 2015*). As there were only three samples from P51M5, this whole patient was excluded from the analyses. In lineages with both over- and non-producers, over-producers were sampled before non-producers in all but three cases. In two cases there was <5 months between sampling, and we scored non-producers as occurring at the same time as overproducers. In one case the non-producer was sampled 2.95 years prior to the over-producer and scored as having occurred in the absence of an over-producer. The subsequent sampling of the over-producer, and an additional non-producer that shared the genotype with the first non-producer, was not included in the analyses (*Appendix 1—figure 1*, P53M4). In two clone types two independent transitions to non-producers were observed (non-producing isolates harboured different pyoverdine mutations) and these were included as independent events (P44F5 and P14M4). We focus here on within-clone type dynamics, as temporal overlap of longitudinally sampled clone types only occurred in two patients (*Appendix 1—figure 1*; *Andersen et al., 2015*).

We categorized each clone type in individual patients as harbouring over-producers or not. If an over-producer was present, the length of time from that sampling to a non-producer was observed, or till the last sample was collected from the patient, was noted. Clone types without non-producers were classified as right-censored. The same was done for patients without over-producers. In total, 45 clonal lineages were followed covering 404 isolates of 33 clonetypes from 35 patients (*Appendix 1—figure 1*; some clone types occur in multiple patients, and some patients are infected by multiple clone types [*Marvig et al., 2015*]). The timing of sampling of non-producers was analysed with the 'survival' package in R, which takes into account the sampling extent of a clonal lineage, that is both the timing and frequency of events (*R Core Team, 2013*; *Therneau, 2015*).

## Identification of mutations in the pyoverdine, pyochelin, *phu*, *has* and *feo* systems

Mutations in the pyoverdine (pvdQ-pvdI +pvdS pvdG+pvcA ptxR+fpvB), the pyochelin (*ampP-pchA*), the *phu* (*prrF1-phuR*), the *has* (*PA3404-hasI*) and the *feo* systems (*PA4357-feoA*) were identified previously from Illumina GAIIx or HiSeq2000 reads (*Andersen et al., 2015*; *Markussen et al., 2014*; *Marvig et al., 2013*; *Marvig et al., 2015*; *Rau et al., 2012*; *Yang et al., 2011*) (*Supplementary file 1C-D*). Mutations of iron systems in the six new DK1 isolates were identified as described previously (*Andersen et al., 2015*), by mapping reads against a DK1 draft genome (*Markussen et al., 2014*). Mapping of reads revealed low sequencing depth of the *hasD* (required for hemophore secretion) and *hasS* (the *has* anti-sigma factor) genes, likely due to repetitive regions. These were unsuccessfully attempted amplified by PCR. Mutations were categorized as synonymous (syn) or non-synonymous (ns) SNPs, insertions or deletions (indels), or intergenic. We tested for a bias in which genes in each operon were mutated with [Poisson test, $P(X \geq \text{changes}_{observed})$~Poisson distribution (pois) $(X; \text{changes}_{expected}) < 0.05$].

We identified the order of mutation across the different iron uptake systems, with a system classified as mutated by the presence of ns SNPs or indels in any gene in the system. For pyoverdine, the presence of mutations was compared to the actual measurements of production; the two measures are highly consistent (*Andersen et al., 2015*) but when incongruent the phenotype was used. For the *phu* region intergenic *phuS*//*phuR* mutations predicted to cause receptor upregulation were included, and when occurring without other *phu* mutations these were recorded separately. Transitions were identified within clonal lineages, that is if an isolate for example had a pyoverdine mutation and measured loss of production, and a subsequent isolate had the same pyoverdine mutation

in addition to a pyochelin mutation, this was considered a transition from wild-type (WT) to loss of pyoverdine, to loss of pyochelin. If for example a *phu* and pyochelin mutation was observed for the first time in the same isolate the exact order could not be inferred. In the transmissible clone type DK2 pyoverdine production was detected to have been lost in an isolate from 1973 and the mutation and phenotype was found in all later isolates. Subsequently, independent clonal lineages for example accumulate *phu* mutations. In the analyses this was counted as one loss of pyoverdine, and then multiple independent *phu* mutations. We calculated the expected order of mutation, equally weighted by the size and number of mutations of the different systems. We tested, as described above, whether this was different from the observed, using only cases where there was a clear order of mutations. The social environment, that is the genotypes of co-occurring isolates, was not considered in this analysis.

For the *phu* system, we located mutations in the *phuR* receptor gene to functional regions (the N-terminal-domain, periplasmic turns, transmembrane strands or extracellular loops) using a predicted structure of the receptor (http://bioinformatics.biol.uoa.gr//PRED-TMBB [*Bagos et al., 2004*]). For each domain in the receptor we calculated the expected number of non-synonymous amino acid changes, as the proportion the domain comprises of the entire protein times the observed number of changes. Whether the observed changes in each domain differed significantly from the expected was calculated as described above.

## Transitions between public and private iron uptake

To test if the appearance of intergenic *phuS//phuR* mutations that cause upregulation of the *phu* system was correlated with overall loss of pyoverdine production in the social environment we categorized each independent acquisition of these mutations as having occurred in the presence or absence of measured pyoverdine production, by the focal isolate and co-occurring isolates. In the absence of pyoverdine production the time between loss of production and occurrence of mutations was estimated. The analysis was performed on longitudinally sampled clonal lineages from both collections of isolates, that is lineages with at least two isolates with or without pyoverdine production. Further, three cases where the loss of pyoverdine production and *phuS//phuR* mutations were observed in the same isolate were also included. Clonal lineages that did not acquire *phuS//phuR* mutations were classified as right-censored. For DK1 and DK2 we used monophyletic clades with and without mutations as independent clonal lineages (Blue lines in *Appendix 1—figure 3* and *4*). Branches with single isolates were not included unless they showed loss of pyoverdine from the social environment or *phuS//phuR* mutations (inclusion of these did not significantly affect the result of the test). We included isolates of DK53 where no WT pyoverdine producer was sampled, but *phuS//phuR* mutations were inferred by comparison to PAO1 (*Supplementary file 1C*).

The *phuS//phuR* mutations only occurred in isolates that had lost pyoverdine production (*Appendix 1—figure 3* and *4*). In three instances these, however, co-occurred in the patient with isolates that produced pyoverdine (*Appendix 1—figure 5* and *6*). The producers and non-producers with *phu* upregulation are unlikely to have interacted at the time when the iron system mutations occurred in two of these. In patient P30M0, infected with clone type DK1, the cooperating isolates were sampled from the patient's sinuses only, while the non-producers with *phu* upregulation came from lung samples. In patient P28F1, two lineages of clone type DK1 co-occurred. One had been transmitted from another patient, where pyoverdine production was lost, and *phu* likely upregulated, prior to transmission and establishment of a co-infection. In patient P82M3 a pyoverdine producer and an isolate with a *phuS//phuR* mutation, both of clone type DK32, were found in the same lung sample. The latter harbours the most common *phuS//phuR* mutation (G 146 upstream//35 upstream A, *Supplementary file 1C*), which is located outside the *phuR* promoter (*Marvig et al., 2014*). It is the only isolate to have this *phuS//phuR* mutation in isolation, all others with it have an additional SNP, and the effect of it is unknown (but see below).

The timing of sampling of mutants was analysed with the 'survival' package in R (*Therneau, 2015*). We are likely to over-estimate the length of time to mutation, as patients chronically infected with the transmissible DK1 and DK2 clone types were infrequently sampled (*Markussen et al., 2014; Marvig et al., 2013*). In these cases, with several years between longitudinal samples, a mutation may only be detected years after occurring.

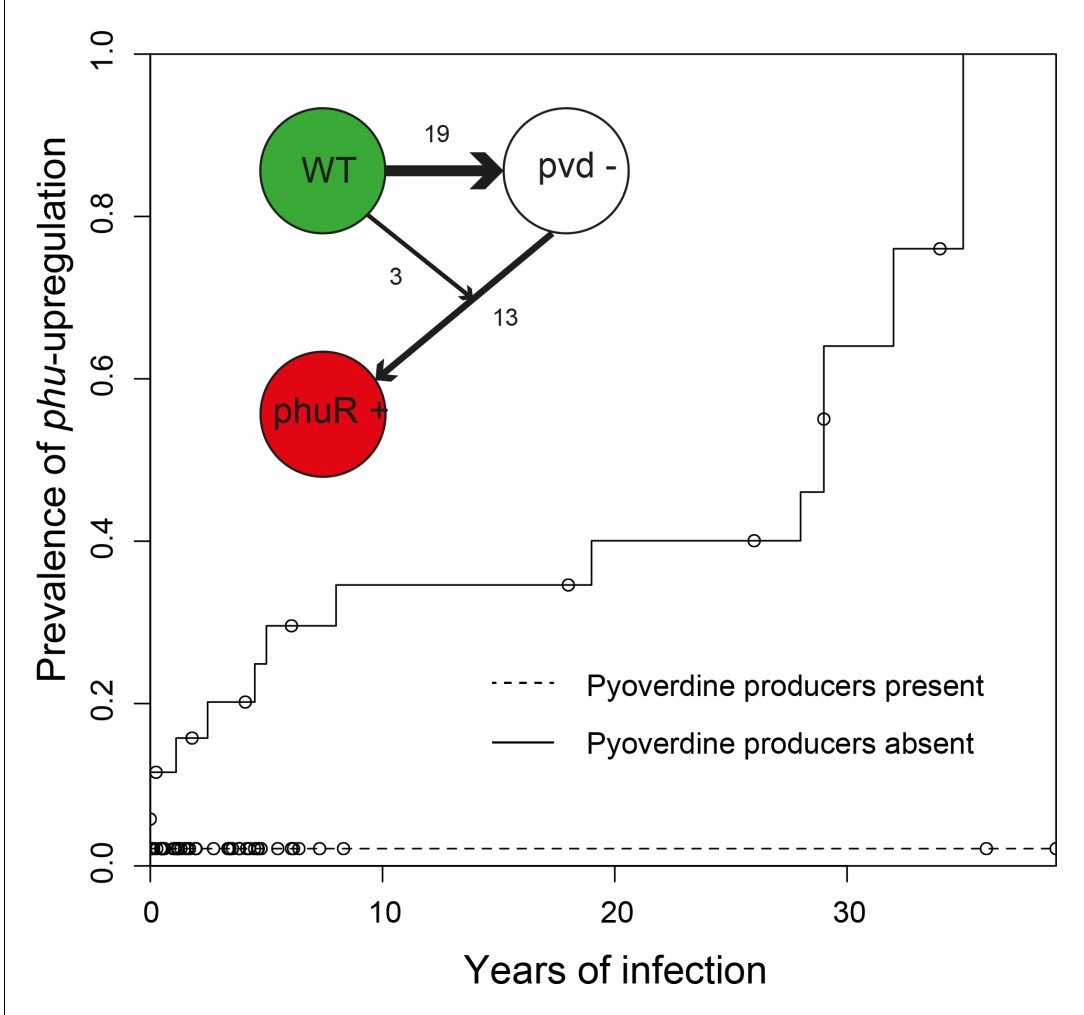

**Figure 3.** Loss of cooperation precedes privatisation of iron uptake. Inverse Kaplan-Meier graph showing that *phuS//phuR* mutations are observed 15 times when pyoverdine is absent from the social environment (solid line, n = 26), and only once when pyoverdine is available in the social environment (dashed line, n = 47, see text). The mutations are only found in clonal lineages that do not produce pyoverdine. Circles indicate when sampling of clonal lineages stopped, without the observation of *phuS//phuR* mutations. Insert shows that a transition from wild-type pyoverdine production (WT) to no production (pvd-) was observed 22 times in clonal lineages, while *phuS//phuR* mutations occurred 16 times independently (phu+), 13 of which were after loss of pyoverdine production from the environment. The order could not be inferred in three cases, in one of which the isolate co-occurred with pyoverdine producers.

DOI: https://doi.org/10.7554/eLife.38594.005

## Uptake of heme following *phuS//phuR* mutations

We tested if the presence of *phuS//phuR* mutations gives a growth advantage in iron-limited media supplemented with heme at different concentrations, using an isogenic pair of DK2 isolates only differing in the presence of a 1 bp *phuS//phuR* deletion (*Marvig et al., 2014*). Three biological replicates of each isolate were cultured in liquid LB media for 24–48 hr, so that all cultures reached an $OD_{600}$ above 0.1. Cultures were subsequently standardised to a starting $OD_{600}$ of 0.001, in iron-limited CAA media following *Kümmerli et al. (2009)*, without heme or supplemented with 1, 2.5, 5 or 10 µM heme as an iron source (BioXtra porcine hemin $\geq$97.0%, Sigma-Aldrich). Of each culture 200 µL were inoculated in 96 well plates. Plates were sealed with a breathable membrane to avoid evaporation and incubated in a plate reader measuring $OD_{600}$ every 30 min for 48 hr. Growth curves were plotted in Excel and the average maximum $OD_{600}$ of the three replicates calculated. The difference in max $OD_{600}$ between the isolates with and without the *phuS//phuR* mutation at the various

heme concentrations was analysed in R with a Two-Way ANOVA test with post hoc comparisons with Tukey HSD (*R Core Team, 2013*).

We also compared growth of pairs of clinical isolates with *phuS//phuR* mutations matched to their closest genetically related isolate without mutations. The *phuS//phuR* mutations occurred in three different clone types without additional *phu* mutations (DK1, DK2 and DK32, highlighted in yellow in *Supplementary file 1C*). Six isolates with mutations were matched with four closest relatives without, because DK2 in two instances had independent *phuS//phuR* mutations in two closely related isolates. The WT isolate for these pairs was used to make the isogenic pair described above, and the mutation of one of these clinical isolates moved to this background (P0M30-1979 and P173-2005; *Appendix 1—figure 7*; *Marvig et al., 2014*). One additional DK2 isolate with a mutation was lost from the collection, and only available as sequence reads (P80F1; *Appendix 1—figure 4* and *Supplementary file 1C*). Growth was measured as described above with 2.5 or 5 µM heme as an iron source, and differences in max $OD_{600}$ within a pair were compared with a t-test in R (*R Core Team, 2013*).

## Results and discussion

### More cooperative populations are more vulnerable to exploitation

The environmental clones that infect CF patients produce pyoverdine (*Andersen et al., 2015*) but what happens as they transition to life in a host? We compared pyoverdine production between isolates sampled at different time points of infection in iron-limited media, in which cells must produce pyoverdine to avoid iron starvation. In some clonal lineages of *P. aeruginosa,* defined as longitudinally sampled isolates of the same clone type that share a recent ancestor, pyoverdine production was maintained for >30 years of CF infection: in the 12 clonal lineages of the transmissible clone type DK1 that were followed for >30 years there were five independent losses of pyoverdine production but three clonal lineages maintained production (*Andersen et al., 2015*); *Appendix 1—figure 3*, and below). Limited diffusion of secreted pyoverdine, as is likely in viscous CF sputum, impairs the ability of potential cheats to exploit and may contribute to stabilize cooperation in these lineages (*Julou et al., 2013*; *Kummerli et al., 2009*); although there may be significant exchange between spatially distinct microcolonies (*Weigert and Kümmerli, 2017*).

When analysing the isolates from 35 frequently sampled young patients of 33 clone types (some clone types infect multiple patients and some patients harbour multiple clone types), we found that pyoverdine production increased in 36% of clonal lineages within the first two years of infection. Overall, these 'super-cooperator' isolates, which had >30% higher production than isolates initiating infection, were detected in 25 of 45 independent clonal lineages (55%; *Figure 1*), and on average sampled 1.89 ± 1.79 st. dev. years into the infection. This suggests that upregulation is initially favoured, likely by iron limitation or inter-species competition. In CF patients, *P. aeruginosa* frequently co-infects with *Staphylococcus aureus* (*Folkesson et al., 2012*), and co-culture in vitro causes an upregulation of pyoverdine production (*Harrison et al., 2008*). When in co-infection, iron sequestering can have the additional benefit of making it unavailable for *S. aureus* or other competitors (*Leinweber et al., 2018*; *Niehus et al., 2017*). Increased production could also be part of a general acute infection phenotype, as either a pleiotropic side-effect or a beneficial trait (*Coggan and Wolfgang, 2012*). For six out of the 25 lineages with super-cooperators we identified candidate mutations in global regulatory and quorum sensing genes that likely cause increased production of pyoverdine as well as other virulence factors, such as pyochelin, pyocyanin and alginate (*Supplementary file 1E*). For the remainder, super-cooperators produced pyoverdine at a consistently higher rate than their ancestors but the genetic basis of upregulation was unclear.

Despite the potential benefits of pyoverdine production, the presence of super-cooperators appears to weaken long-term stability of cooperation in the lung. In 12 out of the 35 patients, we previously observed the appearance of mutants that have lost the ability to synthesise pyoverdine (*Andersen et al., 2015*), sampled on average 3.06 ± 2.29 st. dev. years into infection. Here, we found that the likelihood of these non-producers arising was significantly higher in the presence of super-cooperators [Survival analysis, $\chi^2$ (2, N = 47) = 5.7; p < 0.05; *Figure 1*]. Non-producers were present in four out of 21 clonal lineages without (19%, on average 2.60 ± 2.45 st. dev years into infection) and in eight out of 24 lineages (disregarding one where the non-producer was sampled

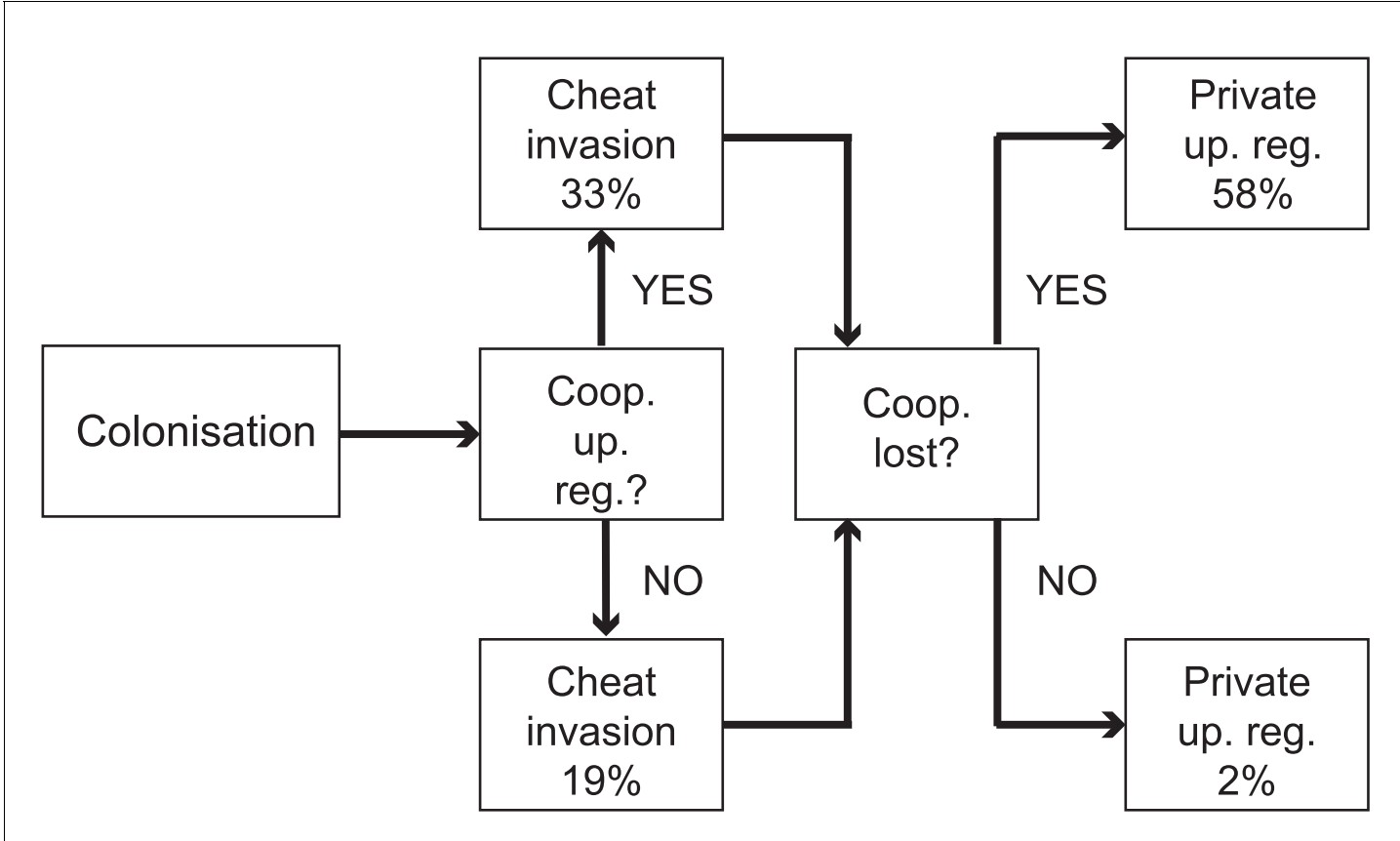

**Figure 4.** Evolutionary trajectories of iron metabolism in the CF airways. Following infection (n = 45 clonal lineages of 33 clone types), pyoverdine production is frequently upregulated (n = 25). Cheaters are significantly more likely to invade following upregulation. Following loss of pyoverdine production in the social environment (n = 26 non-producing clonal lineages of 6 clone types, from eight independent losses of production and two clonal lineages from which no pyoverdine producers were sampled), the *phu* system is frequently upregulated by intergenic *phuS//phuR* mutations (15 times), while this very rarely occurs (one time) when pyoverdine is produced in the social environment (n = 47 producing clonal lineages of 32 clone types).

DOI: https://doi.org/10.7554/eLife.38594.006

prior to the super-cooperator) with super-cooperators (33%, on average 3.32 ± 3.32 st. dev years into infection and 1.45 ± 1.39 st. dev. years after sampling of a super-cooperator). There was no significant difference between the time of sampling of non-producers between lineages with or without super-cooperators (Welch Two Sample t-test, t = −0.54, df = 8.01, p = 0.61). From the data set it is not clear whether cheats evolve from super-cooperators or directly from isolates with WT pyoverdine production, as the genetic basis of over-production is unknown.

Why is upregulation of pyoverdine production associated with the subsequent loss of this trait? Non-producers appear to be favoured as a result of exploiting the pyoverdine production of neighbouring cells – a strategy referred to as cheating (*Ghoul et al., 2014a*). We have previously shown that non-producers in the lung acquire mutations in a pattern consistent with a cheating strategy, as opposed to one expected from redundancy, in that they retained the pyoverdine receptor only if producers were present (*Andersen et al., 2015*). In other words, non-producers lose the capacity to contribute but retain the ability to benefit from pyoverdine, as long as it is being supplied by cooperative neighbours. The relative fitness of cheats is predicted by theory to be greater when there are higher levels of cooperation in a population, because their competitors are bearing a higher cost (*Ross-Gillespie et al., 2007*). This prediction has been shown to hold in empirical tests (*Ghoul et al., 2014b*; *Harrison et al., 2008*; *Jiricny et al., 2010*). Hence, super-cooperators are predicted to be especially vulnerable to exploitation and invasion of cheats. While isolates with a WT level of pyoverdine production may also benefit from the presence of super-cooperators, the larger production

cost differential between non-producers and super-cooperators gives non-producers a greater advantage than WT producers. The pattern of upregulation, followed by loss, is likely to be applicable to the evolution of other exoproducts during infection.

## Evolution of iron acquisition systems in the lung – the sequence of events

How do the changes in cooperative pyoverdine production relate to the overall iron metabolism of the cells? Comparison of the accumulation of mutations across five different iron uptake systems (*Table 1*) and measurement of pyoverdine production showed that the appearance of pyoverdine non-producers is the first change in iron metabolism observed in most clonal lineages, in both isolate collections. Loss-of-function of the cooperative pyoverdine system occurred earliest in 16 out of 25 clonal lineages where the phenotypic measurements and sequence analysis provided a clear order of systems affected (*Figure 2*; *Supplementary file 1F*). These were observed to occur on average 3.06 years after the first sampling of the clone type in the young patients (n = 11 clonal lineages in nine patients), and between 1–39 years after infection for the more rarely sampled transmissible clone types (n = 5 clonal lineages in five patients). When weighed by the high mutation rate of the system, reflecting a strong selection pressure, and its large size, this is not unexpected [Poisson test, $P(X \geq 16) \sim$ pois (X; 14.69) > 0.05], *Figure 2*; *Supplementary file 1D*].

Mutation of the cooperative pyochelin system first was found not to differ from that expected given a random distribution of mutations [Poisson test, $P(X \geq 6) \sim$ pois (X; 5.14) > 0.05], *Figure 2*; *Supplementary file 1D*]. The conditionally private or cooperative *has* heme uptake system was found to acquire few mutations during infection. It remains to be tested if this is because maintenance of the system is favoured by selection, or if selection for loss is relatively weak. The private *feo* system is rarely mutated (*Supplementary file 1D*) and has been suggested to be of increased importance as conditions turn microaerobic during infection (*Hunter et al., 2013*).

Across both isolate collections we observed pyoverdine cheats appear 14 times without going to fixation within the patient during the sampling period, but further recorded eight cases where no producers were sampled a year prior to, or after emergence a non-producer, and two clonal lineages from where no producers are sampled at any point. This is the potential extinction scenario of cheat invasion. However, because we have continued to sample after this event, we can observe how the population responds to this potentially catastrophic situation. One of these events for example occurred in the transmitted clone type DK2 that was subsequently sampled from 16 patients over 35 years. And because cheat invasion is not inevitable, we can compare dynamics in lungs with and without availability of pyoverdine.

## Cheat invasion is associated with a subsequent switch to private iron uptake mechanism

After cooperation was lost from a clonal lineage, and only if it was lost, we observed that iron uptake was privatised (*Figure 3*). This was achieved by upregulation of the private *phu* system. The *phu* system targets the iron-rich heme molecule, which is taken up directly through a membrane-bound receptor PhuR (*Table 1*; *Ochsner et al., 2000*). Increased expression of the *phu* system results from intergenic mutations between the *phuS* gene encoding a heme traffic protein, and the receptor gene *phuR* (*phuS*//*phuR* mutations; *Marvig et al., 2014*). In the isolate collections, a total of 29 SNPs and two indels accumulate in the *phuS*//*phuR* region. Eight of these mutations in the *phuR* promoter have been found to cause a significant upregulation of promoter activity, and one of these has further been shown, in an isogenic pair, to provide a growth benefit to the carrier when heme is available as an iron source (*Marvig et al., 2014*); see also below). These specific mutations were found to arise significantly more frequently than expected by chance following the loss of pyoverdine production [Poisson test, $P(X \geq 5) \sim$ pois (X; 0.39) < 0.01; *Figure 2*]. In three patients, non-producing isolates with *phuS*//*phuR* mutations were sampled at the same time as pyoverdine producers, however in two of these the producer and non-producers were unlikely to interact, and in the third the *phuS*//*phuR* mutation occurred outside the *phu* promoter region with unknown effects on heme uptake (but see *Appendix 1—figure 7*). We thus conclude that *phu* upregulation takes place when pyoverdine production is lost from the social environment. Many of the isolates with *phuS*//*phuR* mutations also harboured pyoverdine receptor mutations that eliminate the possibility

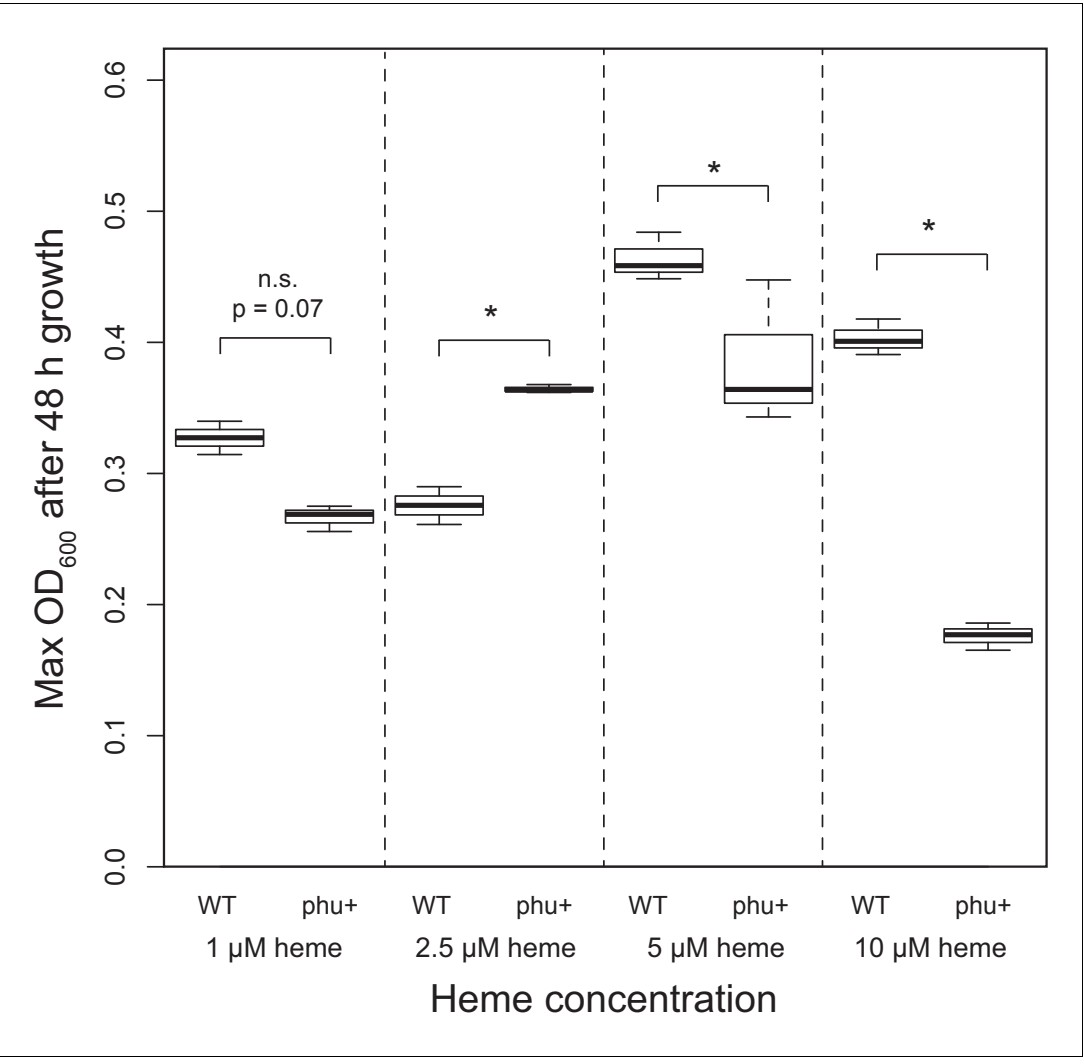

**Figure 5.** Effect of *phu* upregulation depends on the heme concentration. An isogenic pair of isolates differing only in a 1 bp *phuS*//*phuR* deletion was grown in iron-limited media supplemented with heme at 1–10 µM. The clinical isolate without the mutation, that does not produce pyoverdine (WT), had a significant higher maximum $OD_{600}$ after 48 hr growth at 5 and 10 µM, whereas the mutant (phu+) grew best at 2.5 µM. There was no significant difference at 1 µM. Boxplots show median value, and the interquartile range ±25%. The * marks a significant difference at p < 0.05.

DOI: https://doi.org/10.7554/eLife.38594.007

to cheat (6 out of 7 in DK1, 7 out of 7 in DK2, 1 out of 2 in DK53, 0 out of 1 in DK32; *Appendix 1—figure 3–6*; *Andersen et al., 2015*). Knock-out of the pyoverdine receptor is, however, not expected to be a selective force in *phu* upregulation. Rather, the data suggest that both occur because pyoverdine is no longer available in the environment, selecting for privatisation of iron uptake and elimination of the costly receptor (*Andersen et al., 2015*).

Social interactions are unlikely to be the only selection pressure influencing dynamics in iron acquisition strategies. Changes in iron uptake strategy might more readily be expected to reflect changes in the lung environment as the availability of iron in different forms increases with disease progression (*Hunter et al., 2013*). And while iron is found to be equally available in its oxidized ferric ($Fe^{3+}$) and reduced ferrous forms ($Fe^{2+}$) in early infection, there is a skew towards the reduced form later, which the *feo* system targets (*Hunter et al., 2013*). Heme and hemoglobin are also present in the sputum (*Ghio et al., 2013*), and availability has been speculated to increase as the lung tissue degrades (*Marvig et al., 2014*), as evidenced by increased coughing up of blood, termed

hemoptysis, with age (*Thompson et al., 2015*). As such, pyoverdine may be most efficient early in infection, and *phu* only advantageous later as lung tissue degrades and/or alternative sources of iron become more or less abundant. However, the patterns we observe are not consistent with this explanation. If pyoverdine production is maintained, the *phuS//phuR* genotype remains unaltered from colonization, even after more than 30 years of infection (*Figure 3*). The *phuS//phuR* mutations, responsible for upregulating the private *phu* system, are only observed in non-producers, and in 15 out of 16 cases after the loss of cooperation in the population [Survival analysis, $\chi^2$ (2, N = 73) = 11.7; p < 0.01; *Figures 3 and 4*], occurring between 0 to 35 years after the loss of pyoverdine production. Upregulation of private iron uptake can, therefore, be predicted from the social environment, not duration of infection and associated changes in the lung environment.

## Why is private heme uptake only upregulated if cooperation is lost?

The fact that we only observe upregulation of the *phu* system if pyoverdine production has been lost suggests that privatisation is not universally beneficial but may instead represent a final recourse when other mechanisms are no longer functioning. *P. aeruginosa* iron uptake is finely controlled by feedback loops, in response to environmental iron concentrations and need (*Vasil and Ochsner, 1999*). As iron is toxic at high concentrations (*Anzaldi and Skaar, 2010*; *Touati, 2000*), an indiscriminate upregulation of one system could be a significant disadvantage in environments where iron concentrations fluctuate. We, therefore, tested whether the growth benefits of *phu* upregulation (by *phuS//phuR* mutations) were dependent on iron availability, in the form of heme which the *phu* system targets. We examined the growth difference between a clinical isolate that had lost pyoverdine production, and an isolate, isogenic but for a clinical *phuS//phuR* mutation that cause an upregulation of the heme receptor gene *phuR* (*Marvig et al., 2014*). The experiment was performed in iron-limited media supplemented with heme at biologically relevant concentrations, which ranged from <1 µM typical of healthy individuals, across 2.5 µM, 5 µM to 10 µM typical of CF patients (*Ghio et al., 2013*).

Consistent with our hypothesis, we find that *phu* upregulation is only beneficial at a narrow range of intermediate heme concentrations. There was a significant growth difference between the isolates and heme concentrations, and an interaction effect (Two-Way ANOVA; Interaction term: Isolate*-Heme: F = 48.11, df = 3, p < 0.01; *Figure 5*; *Supplementary file 1G*). At 2.5 µM heme the isolate with an upregulated *phu* system achieved a higher density (Tukey HSD, $p_{adj}$ <0.01, *Figure 5*), consistent with previous findings (*Marvig et al., 2014*). In contrast, at 5 and 10 µM the isolate without upregulation had an advantage ($p_{adj}$ <0.05, *Figure 5*). No growth was observed in the absence of heme, and at 1 µM there was no significant difference in growth between the isolates but a trend for the WT to do better ($p_{adj}$ = 0.073). This suggests that *phuS//phuR* mutations lead to increased heme uptake that is beneficial at low external concentrations (>1 and<5 µM heme) but detrimental at high concentrations (>5 µM heme). Analyses of pairs of clinical isolates with and without *phuS//phuR* mutations gave less clear results, but showed a similar pattern of a benefit of *phuS//phuR* mutations at low concentrations of added heme (*Appendix 1—figure 7*). If iron availability increases during infection, an initially beneficial upregulation of heme uptake would turn toxic. In the *phu* system we found a significant bias towards mutations of the *phu* receptor gene *phuR* (Poisson test, P(X $\geq$ 24) ~ pois (X; 10.69) < 0.05; *Supplementary file 1D*). These were not randomly distributed but primarily found in the extracellular loops of the *phu* receptor, that initiate the uptake of heme (*Noinaj et al., 2010*; *Smith et al., 2015*) [Poisson test, P(X $\geq$ 15)~pois (X; 7.28) < 0.05; *Figure 6*]. If there is selection to reduce iron concentration within cells, these may, therefore, represent compensatory mutations.

## Conclusions

Bacterial cells will be under selection to adapt to the host environment, including antibiotic pressure (*Lieberman et al., 2011*). The observations we report here, however, demonstrate the errors of interpretation that can arise from assuming that bacterial cells are driven entirely by the evolutionary imperative to survive in the host. As in all other organisms, fitness is also determined by the ability to compete with members of the same species, and these two evolutionary drivers – host adaptation and competition – do not always coincide. Our results show that the initial upregulation of cooperation, selected by either the environmental conditions, competition or as a pleiotropic side-effect,

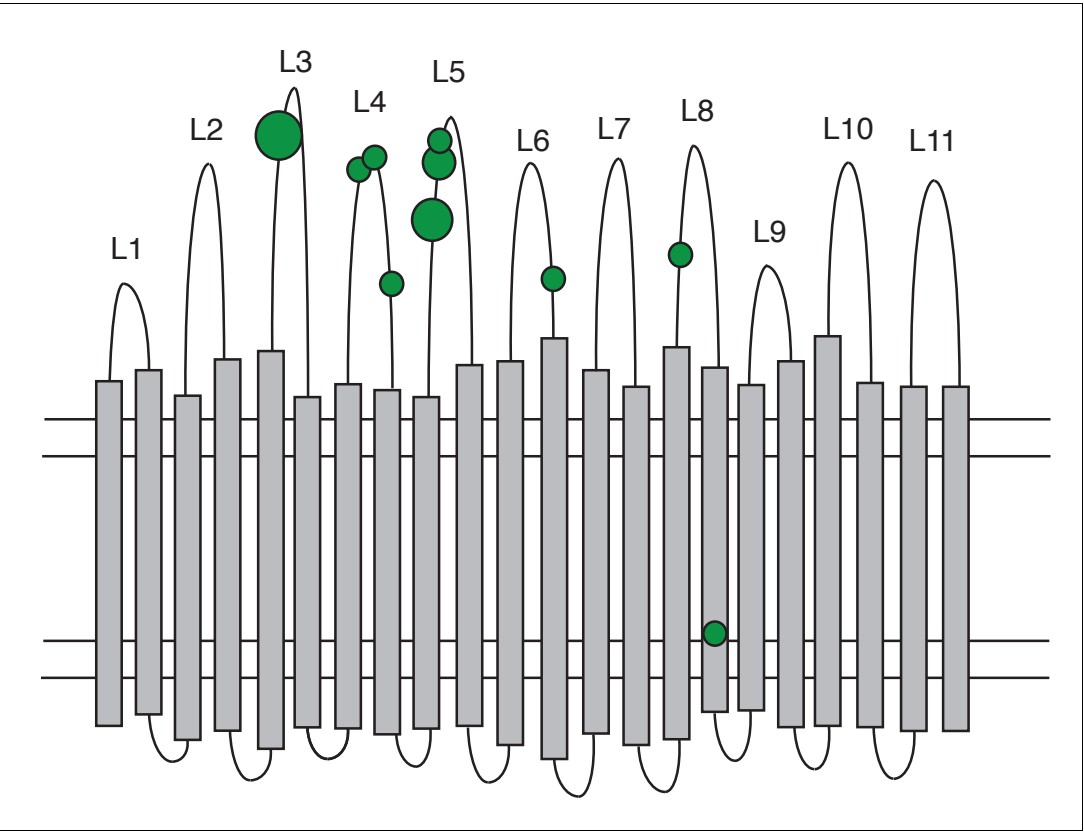

**Figure 6.** Distribution of non-synonymous mutations in the PhuR heme receptor. The figure shows the β-barrel structure spanning the cell membrane (horizontal lines) with transmembrane strands (grey bars), extracellular loops (labelled L1-L11) and periplasmic turns (bottom loops). Green circles indicate the location of non-synonymous SNPs, the smallest circles denoting one mutation and the largest four. Mutations were significantly biased towards the extracellular loops.

DOI: https://doi.org/10.7554/eLife.38594.008

facilitates cheating. The subsequent privatisation of iron uptake is a response to the loss of cooperation, not simply an adaptation to iron availability in the host as previously suggested (*Marvig et al., 2014*), and as we show this carries its own costs. The effect of the social environment on iron acquisition mechanism remains statistically detectable despite the noise from variation in patient age, sex, infection and treatment. In a clinical system like this, failure to appreciate these factors can lead to misinterpretation of how bacterial cells are influenced by the host environment and, therefore, impair necessary understanding for the development of intervention strategies (*Leggett et al., 2014*).

Evolutionary theory clearly predicts that any adaptation involving cooperation can be vulnerable to exploitation, irrespective of its benefits (*Axelrod and Hamilton, 1981*). But this is often overlooked when it is difficult to characterize the selective environment, as for example, inside a human host. By longitudinal sampling of bacterial populations, we were able to observe the consequences of cheat invasion for an essential trait and show that switching to a private strategy can provide bacterial cells with an evolutionary alternative. This observation highlights the importance of considering potential constraints imposed by social interactions: if a private trait is expressed it may not be the most efficient in a given environment, but simply the final recourse in a population with a history of exploitation.

## Acknowledgements

We thank Stuart West, Lars Jelsbak and Rolf Kümmerli for comments on the manuscript. Trine Markussen and Lars Jelsbak provided analysed sequences of additional DK1 isolates, and S M Hossein Khademi provided valuable discussion of the *phu* mutations. This work was supported by grants from Villum Fonden (grant 95-300-13894) and Lundbeckfonden (grant R108-2012-10094) to SBA, the European Research Council (grant SESE) to ASG, and Rigshospitalet (grant R88-A3537), Novo Nordisk Fonden (grant NNF12OC1015920) and Lundbeckfonden (grant R167-2013-15229) to HKJ.

## Additional information

### Funding

| Funder | Grant reference number | Author |
| --- | --- | --- |
| Villum Fonden | 95-300-13894 | Sandra Breum Andersen |
| Lundbeckfonden | r108-a10094 | Sandra Breum Andersen |
| Danmarks Grundforsknings-fond | 126 | Rasmus L Marvig |
| Rigshospitalet | R88-A3537 | Helle Krogh Johansen |
| Novo Nordisk | NNF12OC1015920 | Helle Krogh Johansen |
| Lundbeckfonden | R167-2013-15229 | Helle Krogh Johansen |
| H2020 European Research Council | SESE | Ashleigh S Griffin |

The funders had no role in study design, data collection and interpretation, or the decision to submit the work for publication.

### Author contributions

Sandra Breum Andersen, Conceptualization, Data curation, Formal analysis, Funding acquisition, Validation, Investigation, Visualization, Methodology, Writing—original draft, Project administration, Writing—review and editing; Melanie Ghoul, Investigation, Methodology, Writing—review and editing; Rasmus L Marvig, Formal analysis, Writing—review and editing; Zhuo-Bin Lee, Investigation; Søren Molin, Helle Krogh Johansen, Resources, Funding acquisition, Writing—review and editing; Ashleigh S Griffin, Conceptualization, Supervision, Funding acquisition, Methodology, Writing—original draft, Project administration, Writing—review and editing

### Author ORCIDs

Sandra Breum Andersen http://orcid.org/0000-0002-7030-040X
Rasmus L Marvig http://orcid.org/0000-0002-5267-3173
Helle Krogh Johansen http://orcid.org/0000-0003-0268-3717

### Ethics

Human subjects: The CF isolates used for sequencing were obtained from the Department of Clinical Microbiology, Rigshospitalet, Copenhagen, where they had been isolated from CF patients treated at the Copenhagen CF clinic. The use of isolates was approved by the local ethics committee at the Capital Region of Denmark Region Hovedstaden: registration number H-1-2013-032. All patients have given informed consent. In patients under 18 years of age, informed consent was obtained from their parents. All patient isolates were anonymized prior to any experimental use or analysis.

### Decision letter and Author response

Decision letter https://doi.org/10.7554/eLife.38594.032
Author response https://doi.org/10.7554/eLife.38594.033

## Additional files

### Supplementary files

• Supplementary file 1. Tables with Supplementary Information. (**A**) Calculation of estimated generation times of infecting clone types. For each, the patient ID, clone type, length of infection in years and minutes, estimated doubling time and calculated number of generations is given. A doubling time of 74 min was used for clonal lineages infecting for less than 6 years, 83 min for 6–24 years and 109 min for >24 years based on the measurements of *Yang et al. (2011)*. This is, to our knowledge, the best available for CF conditions. (**B**) Overview of patients and isolates. For each patient is given the ID, the number of samples, the year of first sample, the extent of sampling in years, whether pyoverdine non-producers were sampled, the number of clone types sampled, and the clone type name(s). (**C**) List of mutations in genes associated with iron metabolism. For each isolate, the patient ID, clone type and mutations are listed. A //denotes an intergenic mutation. The isolate names refer to previous descriptions (*Andersen et al., 2015*; *Markussen et al., 2014*; *Marvig et al., 2015*). The six DK1 isolates that were not included in the previous analysis of pyoverdine mutations had one ns SNP in pvdL (A5702G) and one in pvdP (C1508T, patient P55M3, isolate TM50 and TM51). The six isolates used in the phenotypic assay of *phuS//phuR* mutations are highlighted in yellow. (**D**) Summary of the mutations found in five different iron uptake systems in *P. aeruginosa*. The size of the respective systems is given in kb, and mutations are listed as non-synonymous (ns) SNPs, synonymous (syn) SNPs, insertions and deletions (indels) and intergenic (IG). The overall mutations, and ns SNPS and indels, per kb are listed. The *phu* operon experienced the highest mutation rate, dominated by intergenic mutations. Two genes in the *has* operon were poorly sequenced in the majority of isolates, and this is taken into account in the size and mutations listed in parentheses. (**E**) Mutations identified in pyoverdine over-producers that may contribute to this phenotype. For each mutation, the patient ID, clone type and gene is listed, as well as a reference to previous work showing a link between the gene and pyoverdine production (*Frangipani et al., 2014*; *Ochsner et al., 2002*; *Stintzi et al., 1998*; *Visaggio et al., 2015*; *Wu et al., 2004*). (**F**) Transitions between iron uptake systems characterized as the order in which mutations accumulate. When a clear order of mutations could be established the observed and expected number of transitions, given a Poisson distribution, is stated and whether the difference between them is statistically significant. (**G**) Results of Tukey HSD post hoc comparison of a Two-Way ANOVA analysing the difference in Max $OD_{600}$ between an isogenic pair of isolates differing in a *phuS//phuR* mutation at four different heme concentrations.
DOI: https://doi.org/10.7554/eLife.38594.009

• Transparent reporting form
DOI: https://doi.org/10.7554/eLife.38594.010

### Data availability

Pyoverdine reads are available on Dryad at DOI: https://doi.org/10.5061/dryad.6963pj3, and were used to identify isolates with increased pyoverdine production, as presented in Appendix 1-Fig. 1 and Fig. 1. Sequencing data is previously published and available from GenBank. The mutation lists for the isolates from the Children's collection from Marvig et al., 2015, can be downloaded from https://www.nature.com/articles/ng.3148#supplementary-information. The mutation list for the transmissible clone type DK1 from Markussen et al., 2014, can be downloaded from https://mbio.asm.org/content/mbio/5/5/e01592-14/DC1/embed/inline-supplementary-material-1.pdf. The mutation lists for the transmissible clone type DK2 from Marvig et al., 2013, can be downloaded from https://doi.org/10.1371/journal.pgen.1003741.s009 and https://doi.org/10.1371/journal.pgen.1003741.s010. Datasets S03 and S04 from Andersen et al, 2015, with lists of mutations in the pyoverdine system, are downloadable from http://www.pnas.org/content/112/34/10756/tab-figures-data#-fig-data-tables. The raw reads and mutation lists were used to identify mutations in five iron uptake systems, as presented in Supplementary Table 1C and D. Table S03 from Yang et al., 2011, is downloadable from http://www.pnas.org/content/108/18/7481/tab-figures-data, and was used to estimate the number of generations that Pseudomonas aeruginosa was followed for in cystic fibrosis patients.

The following dataset was generated:

| | Database and |
|---|---|

| Author(s) | Year | Dataset title | Dataset URL | Identifier |
|---|---|---|---|---|
| Andersen SB, Ghoul M, Marvig RL, Lee Z, Molin S, Johansen HK, Griffin A | 2018 | Data from: Privatisation rescues essential function following loss of cooperation | https://dx.doi.org/10.5061/dryad.6963pj3 | Dryad Digital Repository, 10.5061/dryad.6963pj3 |

The following previously published datasets were used:

| Author(s) | Year | Dataset title | Dataset URL | Database and Identifier |
|---|---|---|---|---|
| Marvig RL, Sommer LM, Molin S, Johansen HK | 2015 | Convergent evolution and adaptation of Pseudomonas aeruginosa within patients with cystic fibrosis | https://www.ncbi.nlm.nih.gov/sra?term=ERP004853 | Sequence Read Archive, ERP004853 |
| Rau MH, Marvig RL, Ehrlich GD, Molin S, Jelsbak L | 2010 | Deletion and acquisition of genomic content duringearly stage adaptation of Pseudomonas aeruginosa toa human host environment | http://www.ncbi.nlm.nih.gov/nuccore/CP003149.1 | Nucleotide, CP003149.1 |
| Andersen SB, Marvig RL, Molin S, Johansen HK, Griffin AS | 2015 | Long-term social dynamics drive loss of function in pathogenic bacteria | https://www.ncbi.nlm.nih.gov/bioproject/300976 | BioProject, PRJEB8028 |
| Marvig RL, Johansen HK, Molin S, Jelsbak L | 2013 | Genome analysis of a transmissible lineage of Pseudomonas aeruginosa reveals pathoadaptive mutations and distinct evolutionary paths of hypermutators | https://www.ncbi.nlm.nih.gov/sra/?term=ERP002277 | Sequence Read Archive, ERP002277 |
| Rau MH, Marvig RL, Ehrlich GD, Molin S, Jelsbak L | 2010 | Pseudomonas aeruginosa PADK2_CF510, whole genome shotgun sequencing project | https://www.ncbi.nlm.nih.gov/nuccore/AJHI00000000.1/ | Nucleotide, AJHI00000000 |

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

# Appendix 1

DOI: https://doi.org/10.7554/eLife.38594.011

## The sampling regimen behind the two isolate collections.

We used two collections of *P. aeruginosa* isolates to study iron metabolism during CF infection. For the 'children's' collection the young patients have been sampled frequently, for up to 10 years, and that the majority are infected with unique environmental clone types. When *P. aeruginosa* was found to transmit between CF patients (*Cheng et al., 1996*; *Jelsbak et al., 2007*), care was taken to avoid that younger patients become infected with adapted clone types in the clinic. In the collections we use this is clear as the children's collection encompass 54 clone types while the collection of isolates from adults contain only two, highly transmissible, clone types. This collection covers >30 years. To give an overview of the sampling regimen we present first an overview of isolates sampled from the individual patients (*Appendix 1—figures 1* and *2*). To further illustrate how the DK1 and DK2 evolve and transmit between patients we present phylogenies of the two transmissible clone types with the primary changes in iron metabolism in the pyoverdine and *phu* systems highlighted (*Appendix 1—figures 3* and *4*).

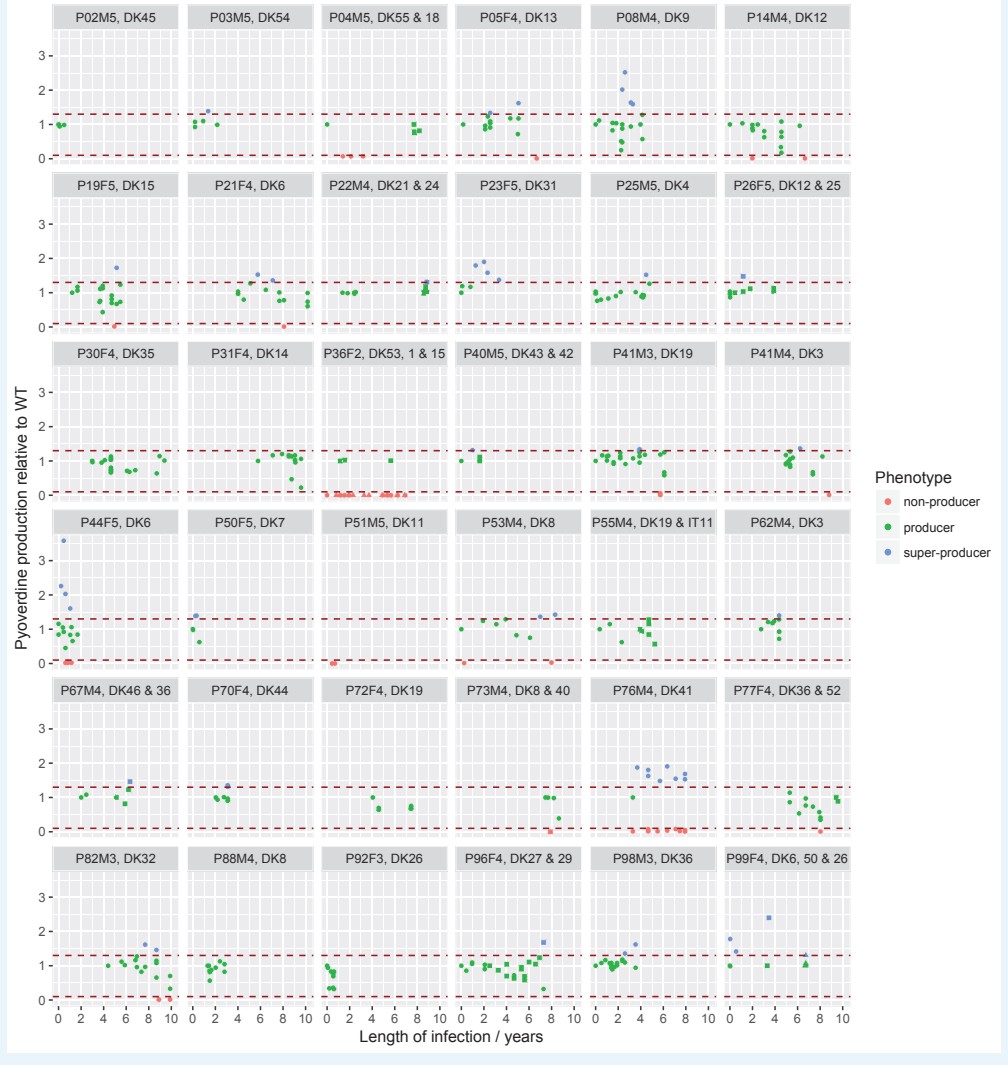

**Appendix 1—figure 1.** Sampling regimen and standardised pyoverdine production of

longitudinally sampled clonal lineages (>1 isolate) from the children's collection. Plots show the sampling time of isolates from individual patients denoted by the patient code and clone type. When more than one clone type was sampled they are ordered so that the first listed corresponds to the dot symbol, followed by the square and triangle. The y-axis shows pyoverdine production standardized by production of the first sampled isolate(s). Isolates with a production of 10–129% of the WT are green, non-producers with a production at <10% of the WT are red, and super-cooperators with a production >130% of the WT are blue. Dashed red lines mark these cut-off points. Four clonal lineages of a 'chronic' phenotype with no pyoverdine producers sampled were excluded from the analysis on super-cooperators (DK1 and DK53 in P36F2, DK40 in P73M4, and DK11 in P51M5). When sampling is shown to start later than year 0 (as e.g. for P67M4) there was a sample collected earlier from the patient that either was not sequenced or of a clone type only sampled once. Created with ggplot2 in R (*R Core Team, 2013*; *Wickham, 2016*).

DOI: https://doi.org/10.7554/eLife.38594.012

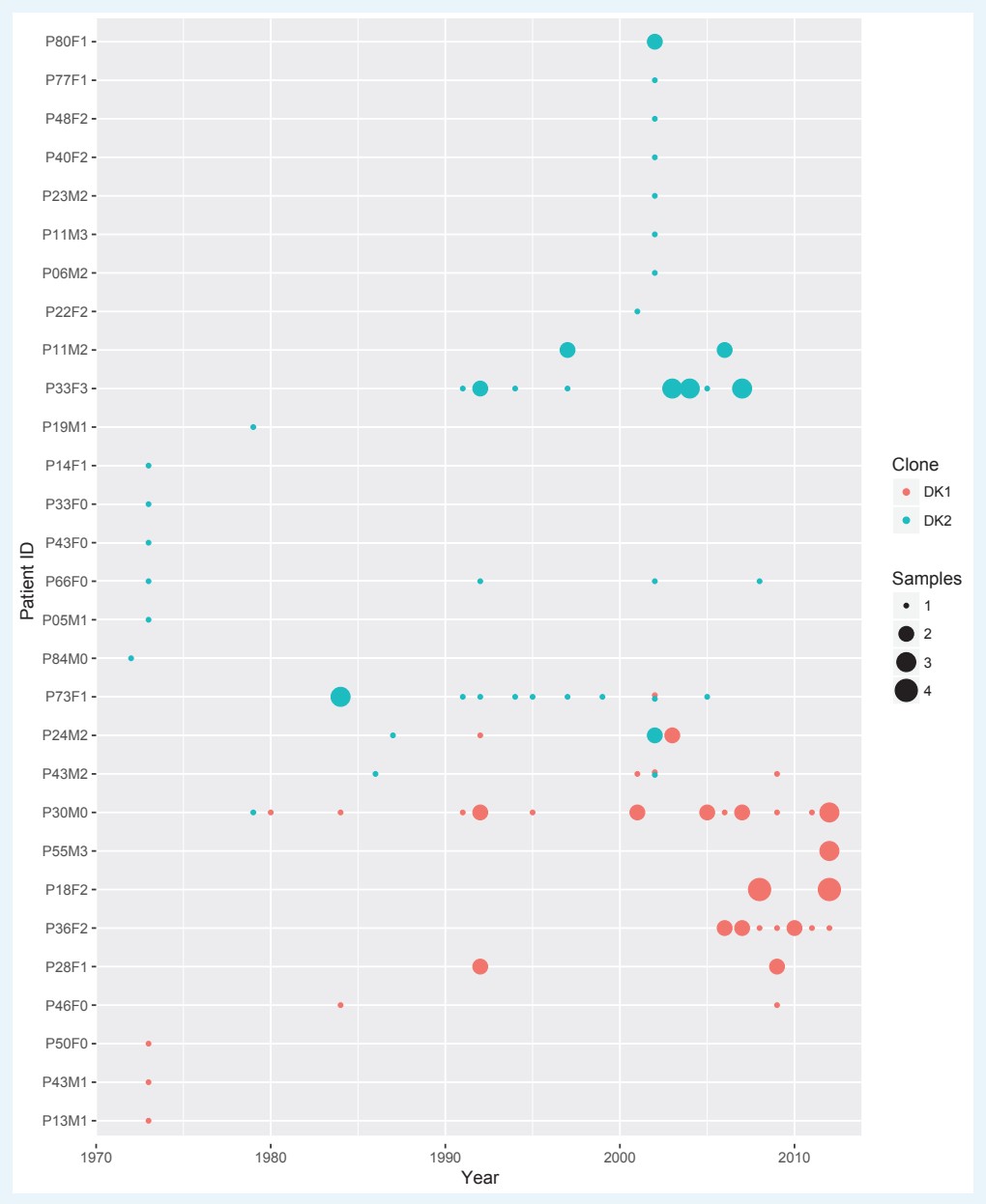

**Appendix 1—figure 2.** Sampling regimen for the transmissible clone types DK1 and DK2. On

the x-axis is sampling year and on the y-axis the patient ID. Red dots show DK1 and blue DK2. Dot size reflects the number of samples collected in a given year (1-4). Note that patient P36F2 from the children's collection is included here as this patient harboured DK1 in addition to two other clone types (see also *Appendix 1—figures 1* and *6*). Created with ggplot2 in R (*R Core Team, 2013*; *Wickham, 2016*).

DOI: https://doi.org/10.7554/eLife.38594.013

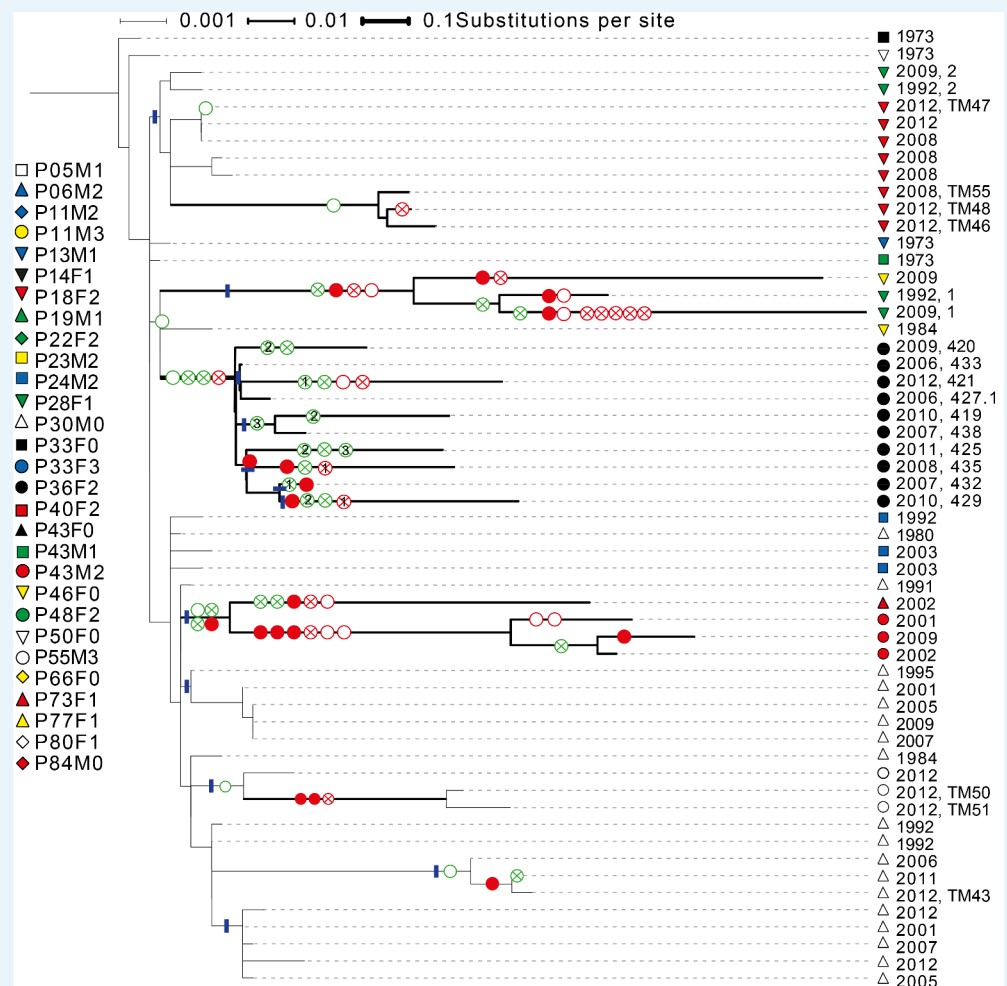

**Appendix 1—figure 3.** Phylogenetic tree of DK1 samples. Tree was generated as described in Methods. Patient ID is denoted by shape and sampling year is listed. When multiple samples were collected from the same patient the same year, they are labelled corresponding to labels used in *Supplementary file 1C*. Scale depends on branch thickness; the thickest branch is only used for samples from P36F2. A green empty circle shows the loss of pyoverdine production measured phenotypically. A green circle with a cross shows the occurrence of a non-synonymous mutation in the pyoverdine receptor gene *fpvA* or the receptor sigma-factor gene *fpvI*. A full red circle shows *phuS*//*phuR* mutations, a red circle with a cross non-synonymous mutations in the *phu* receptor gene *phuR* and an empty red circle additional non-synonymous *phu* mutations. Additional mutations in the pyoverdine, pyochelin, *has* and *feo* systems are not shown. Blue lines mark monophyletic clades used in the analysis on *phu* upregulation. For isolates from P36F2, one *phuR* mutation, and three in *fpvA* were found in multiple isolates on polyphyletic branches (marked with 1, 2 and 3). These are likely not independent, and were not treated as such. Differences in pyoverdine mutations between isolates from P36F2 and P46F0 and P28F1 suggest two independent losses of production..

DOI: https://doi.org/10.7554/eLife.38594.014

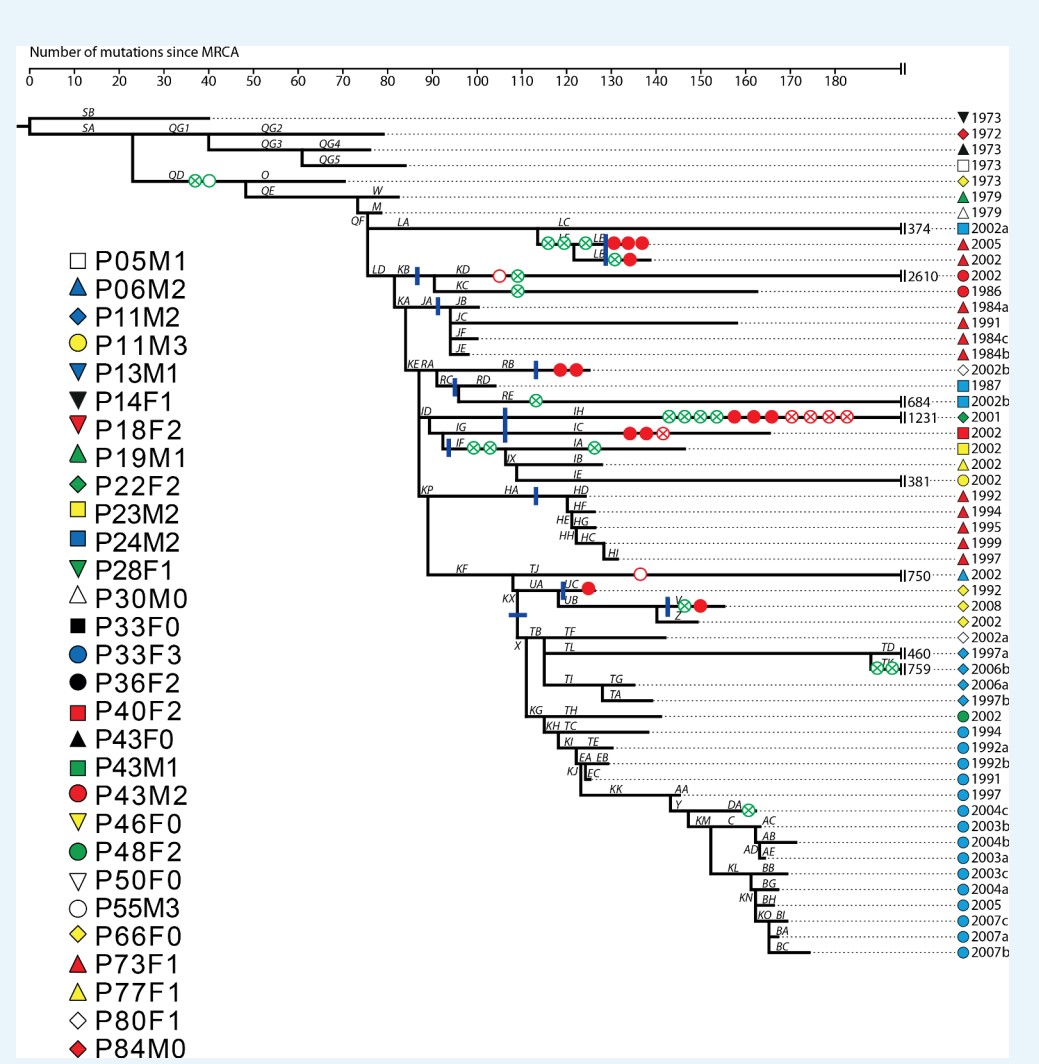

**Appendix 1—figure 4.** Phylogenetic tree of DK2 samples, adapted from (*Marvig et al., 2013*). Pheno- and genotypes highlighted as in *Appendix 1—figure 3*.

DOI: https://doi.org/10.7554/eLife.38594.015

To clarify the within-patient dynamics and social environment we provide an overview of the phenotypes of isolates for the 11 patients with DK1 and DK2 that were sampled longitudinally (*Appendix 1—figure 5*). There are two patients (P28F1 and P30F0) where DK1 isolates that have lost pyoverdine production and have *phu* upregulated are sampled concurrently with WT pyoverdine producers. In P30F0 the isolates with *phu* upregulation were isolated from the sinuses, and the rest from the lungs, and these have therefore inhabited different niches. In P28F1, the pyoverdine and *phu* mutations occurred in another patient (likely P46F0, *Appendix 1—figure 3*) and were transmitted to patient P28F1 where another DK1 lineage also established. In three patients DK1 and DK2 overlapped in time (P24M2, P43M2 and P73F1, *Appendix 1—figure 5*). Could the two clone types have interacted, in regards to iron uptake, in these three patients? Patient P73F1 was first found to be colonized by DK2 in 1984, which had already lost pyoverdine production and uptake ability in 1973 (*Appendix 1—figure 4*). In the last two isolates there were two independent upregulations of the *phu* system (*Appendix 1-fig. 4* and *5*). A DK1 isolate was sampled in 2002 that also had pyoverdine production and uptake knocked-out, and carried *phuS//phuR* mutations for *phu* upregulation. From the DK1 phylogeny (*Appendix 1—figure 3*) it is unclear whether these mutations in DK1 occurred in patient P73F1 and were transmitted to patient P43M2 or vice versa. Given the simultaneous upregulation of *phu* in DK2 in P73F1 we speculate that they

may have occurred in this patient. In patient P24M2 a WT-like DK1 clonal lineage is sampled over a timespan of nine years co-occurring with the non-cooperative and non-cheating DK2. The young patient P82M3 was infected by clone type DK32, and three of the late isolates had lost pyoverdine production (*Appendix 1—figure 6*). Two of these, co-occurring with pyoverdine producers, had a *phuS//phuR* mutation outside of the *phuR* promoter, as discussed in the main text. Patient P36F2 was first infected by DK53 and DK1, neither of which produced pyoverdine and DK53 further had *phuS//phuR* mutations. A WT- like pyoverdine producer, DK15, was later sampled (*Appendix 1—figures 1* and *6*).

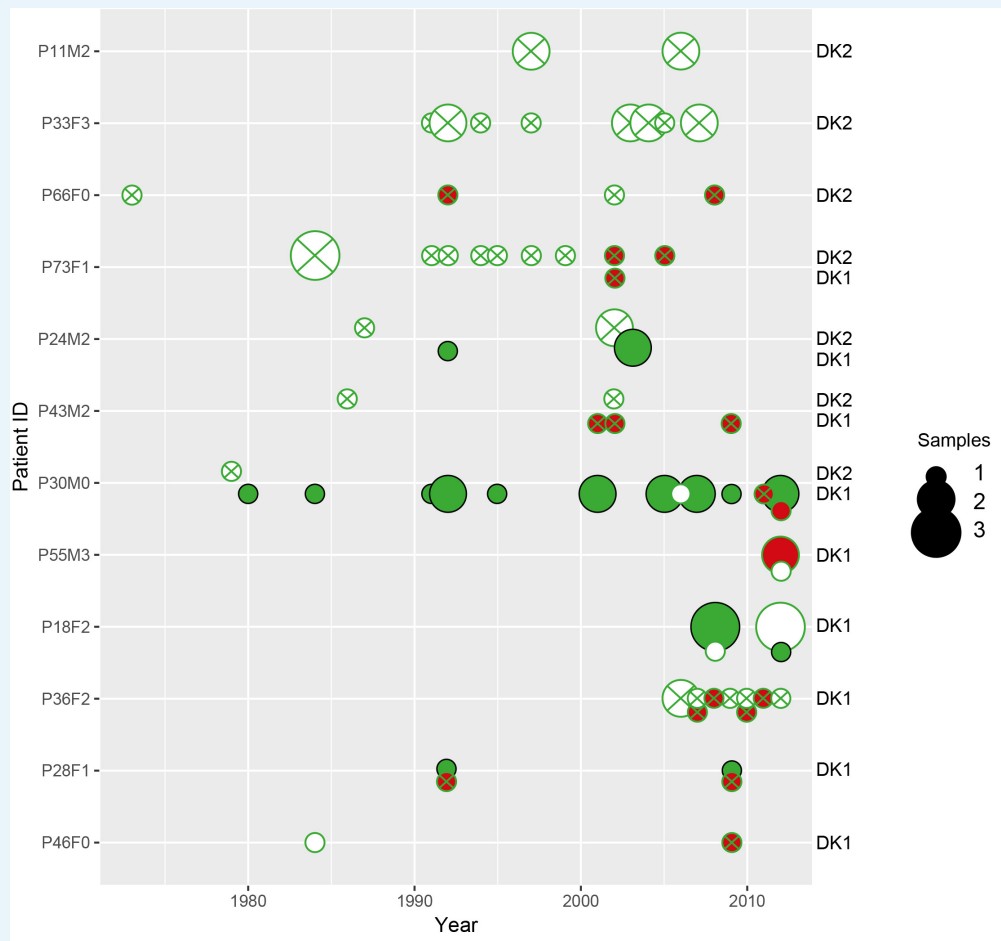

**Appendix 1—figure 5.** Phenotypes of longitudinally sampled DK1 and DK2 lineages. Size of circle denotes the number of samples (1-3). WT-like isolates that produce pyoverdine are shown as a full green circle, non-producers in white with a green stroke, pyoverdine receptor mutations with a green cross and *phu*-upregulation by *phuS//phuR* mutations is shown with red. Created with ggplot2 in R (*R Core Team, 2013*; *Wickham, 2016*).

DOI: https://doi.org/10.7554/eLife.38594.016

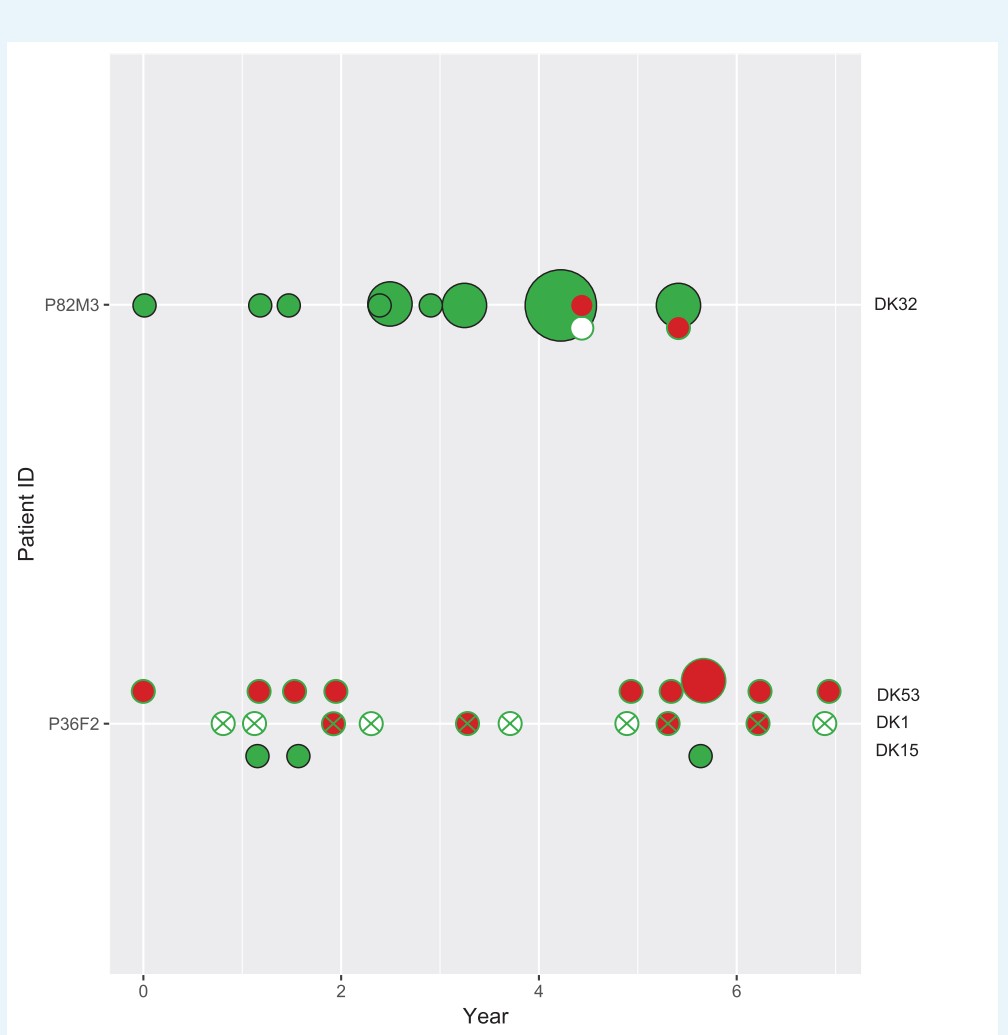

**Appendix 1—figure 6.** Phenotypes of co-occurring clone types in two patients from the children's collection. Phenotypes as in *Appendix 1—figure 5*. Size of circle denotes the number of samples (1, 2 and 4). On the x-axis years since colonization. Note that patient P36F2 is shown in both *Appendix 1—figures 5* and *6*.

DOI: https://doi.org/10.7554/eLife.38594.017

## Growth effect of *phuS//phuR* mutations in clinical isolates.

We tested if the *phuS//phuR* mutations are beneficial in the clinical isolates, using six isolates of three different clone types that had acquired a variety of *phuS//phuR* mutations, and no additional *phu* mutations (*Supplementary file 1C*, marked in yellow). We compared fitness of the mutants with their closest genetically related isolate without *phuS//phuR* mutations. The isolates in a pair were, however, separated by numerous other mutations as they were sampled up to 26 years apart. The pairs were grown in iron-limited media with 2.5 and 5 μM heme added. Three mutants with intergenic SNPs had significantly higher (P73F1; 26% higher; t = −6.33, df = 3.48, p > 0.01 and P83M2; 16% higher, t = −4.48, df = 2.15, p < 0.05; isolates sampled 0 and 23 years apart) or a non-significant tendency towards a higher maximum $OD_{600}$ in 5 μM heme compared to their closest relative (P66F0; 17% higher, t = −2.85, df = 2.67, p = 0.07; isolates sampled 10 years apart; *Appendix 1—figure 7*). A mutant with a three bp insertion, and an isolate with a one base pair deletion in *phuS//phuR* in addition to a SNP, showed no significant difference in growth compared to their closest relative (17% higher,

t = −1.78, df = 2.95, p = 0.18; 16% lower, t = - 0.55, df = 2.56, p = 0.10; isolates sampled 3 and 26 years apart; *Appendix 1—figure 7*). These two isolates have, however, previously been shown to have respectively 25 and 116 fold increased expression of the *phu* receptor compared to the wild type (*Marvig et al., 2014*; *Yang et al., 2011*), and transfer of the one base pair deletion to its ancestor confirms that it infers growth benefits in heme media (*Marvig et al., 2014*; *Figure 5*). This suggest that other mutations across the genome affect the phenotype we are interested in, which is not unexpected given that we are comparing isolates sampled years apart, and the complex genome of *P. aeruginosa* where almost 10% of the genes are predicted to be regulatory (*Stover et al., 2000*). There were no significant growth differences with 2.5 µM heme added. Both isolates of the sixth pair, with a SNP, grew very slowly in rich LB media, and not at all in iron-limited media with or without heme. It is not clear why the beneficial effect of heme in these isolates is seen at a higher concentration compared to the isogenic pair (5 µM vs 2.5 µM, *Figure 5*).

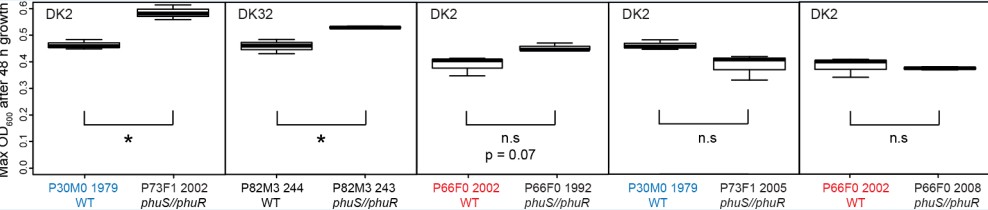

**Appendix 1—figure 7.** Growth of pairs of clinical isolates with and without *phuS*//*phuR* mutations. Isolates were grown in iron-limited media with 5 µM heme added. Pairs where the same WT isolate is used because of two independent acquisitions of *phuS*//*phuR* mutations are highlighted in red and blue. Boxplots show median value of three biological replicates, and the interquartile range ±25%. * marks a significant difference. Plotted in R (*R Core Team, 2013*; *Wickham, 2016*).

DOI: https://doi.org/10.7554/eLife.38594.018

