## [Decision Letter]

Thank you for submitting your article "Privatisation rescues essential function following loss of cooperation" for consideration by *eLife*. Your article has been reviewed by three peer reviewers, and the evaluation has been overseen by a Reviewing Editor and Ian Baldwin as the Senior Editor. The following individual involved in review of your submission has agreed to reveal their identity: Angela Wilks (Reviewer #3).

The reviewers have discussed the reviews with one another and the Reviewing Editor has drafted this decision to help you prepare a revised submission.

Summary:

This manuscript by Andersen et al. describes the evolution of *Pseudomonas aeruginosa* in the lungs of people with CF with a particular focus on evolutionary contingency around cooperative iron acquisition systems. The authors analyze isolates obtained from CF patients sputum in two cohorts of patients with longitudinal sampling and conclude that a series of invasions caused by social evolution take place in the CF lung: first, social cheats invade in siderophore (public good) producing populations, leading to the demise of public good producers. Surprisingly, this event is followed by the raise in frequency of private iron acquisition strategies, which should in principle be able to compete against cheaters. This type evolutionary dynamics, reminiscent of a rock-paper-scissors game, are fundamentally interesting and potentially relevant in the context of pathogenesis, as they could shed light on the selective pressures that impact the evolution of *Pseudomonas aeruginosa* in CF patients.

All reviewers felt this was a potentially interesting and exciting study, however, a number of serious issues were raised by the reviewers (not by reviewer #3, who was brief but positive). These issues require your careful consideration before publication. Many of them revolve around the presentation of the data, which we felt lacked the required details. This lack of detail led to confusion by some of the reviewers, as you will see in their questions below. To be able to consider this paper for publication, we strongly encourage the authors to provide a better overview of the raw data in the main text (e.g. number of isolates, origin, time of isolation, numbers of isolates per patient, phenotypes tested, etc.). This could be done possibly through a dedicated figure or table.

We also found the style of writing to often be too colloquial. Specifically, frequent use of phrases like "game over", "dead end", "last resort" obscured the message of the paper. Other examples include "Life, however, goes on" and there are more. Along the same lines, the Abstract seems to sacrifice clarity for "impact". This needs to be corrected. See comment 6.

One of the reviewers also questioned the novelty of the results, relative to previous findings by the same group. The authors need to clarify their main new data relative to previous work.

We expect to see a detailed point by point reply to the comments you will read below. Given that many of the questions were related to a lack of clear scientific presentation, once we receive your comments we will have to reevaluate the novelty and significance of the findings.

Essential revisions:

1) Clarity of the data: It is very hard to follow the numbers in this work! Numbers are differently represented between figures and text and often do not seem to match. For example, In Figure 1 and the relevant text they claim they observed 14 pvd- cheating events (9+5), while later on they claim they had 23 events (14+9) (subsection “Evolution of iron acquisition systems in the lung – the sequence of events”). What is the relation between these two numbers? The authors need to find a better way to represent the data in a clearer fashion. We strongly suggest to have a new figure with a visualization of the number of strains per time point, per patient and their phenotypes/genotypes.

2) Relation to previous work by the same group: The main problem I see here is the relation between this work and the mBio paper by Marvig et al., 2014, and the PNAS paper by Andersen et al., 2015, both done by the same research groups involved in this work and the latter with the same first author as the current work. The authors often refer to these two papers in the current manuscript, but reading these two papers reveal that most of the basic observations mentioned in the current text were already made in these two papers. It is therefore unclear what is the novelty of this manuscript.

a) Cheating on pyoverdine production: Andersen et al., 2015, have shown that pvd mutations first arise in pvd production pathway (mainly pvdS) and later in the reception pathway, only once the pvd- production mutants are fixed in the population (based on other strains obtained in the same sample and in previous samples of the same patient). They hypothesize that production mutants are cheaters and that pvd reception mutants are favorable due to the cost of reception in the absence of pvd-related benefit. Several points should be made with respect to this manuscript.

i) First, it seems that the basic observation of cheating on pvd production has already been made. The only thing that is added in this manuscript is the observation regarding the relation between cheating and super-producers, to which I have several reservations (see separate point below).

ii) In the current work, only production mutants are discussed, while reception mutants are not mentioned at all. Since both pvd reception mutants and phu+ mutants are correlated with fixation of the pvd production mutants, it is surprising that reception mutants were not discussed and compared with phu+ mutants.

b) Marvig et al., has already identified the phu+ promoter mutation and the fact that they arise only in pvd production mutants, though their work was done on a smaller number of lineages. Therefore, it is not clear what is the added value of the current work? Is it the demonstration that phu+ alleles are only beneficial at low heme levels (see comments on the experiments in a separate point)? Is the claim for novelty in this work depends on the enlarged statistical power? Or is it the claim that phu+ promoter mutants arise only when pvd+ phenotype is completely eliminated from the social environment and not only from the specific lineage? See comments on sociality below for the last option.

Altogether, the distinction between the current work and previous works of the group are not sufficiently clear.

3) Social interaction: It is hard to understand when do the authors discuss social environment within the given lineage (i.e., when discussing interaction between a mutant and its ancestor) and when do they discuss social environment based on multiple strains sampled from the same patient at the same time. While the authors claim social explanations for their observation, much of the analysis in the text is not based on direct social observations, but on comparisons between a mutant and its ancestors (e.g., Figure 1, 2, 3). The two places where a social context is mentioned are:

a) When they discuss the social environment of pvd- cheats, the authors mention that 14 events are not fixed while 9 other are fixed (subsection “Evolution of iron acquisition systems in the lung – the sequence of events”).

b) Later, the authors mention that the phu+ mutations occurred significantly more when Pvd was lost, but here it is unclear whether they mean that Pvd was lost from the genome of the strains, or from the total social environment (subsection “Cheat invasion is associated with a subsequent switch to private iron uptake mechanism”), as they refer to Figure 2 which, to the best of my understanding, discussed only changes within a given genome and not the social context. On the other hand, they specifically mention that in all but three cases (I guess therefore that in 31-3=28 cases, but again – distinction between lineages and patients is unclear), phu+ arise when no pvd+ strains were observed at all.

In both cases there is insufficient information given on the social environment – how many strains were analyzed in any given sample. This data in neither presented nor analyzed in anyway whatsoever in the text. Now, this is obviously important – I would find the claim for fixation of a given genotype (or the elimination of another) to be much stronger if 10 isolates were sampled from the same patient at the same time than if only two samples were taken (or one!). Yet, the social data is completely absent, so I do not know how to interpret the claim for fixation of the cheater. As this is, to the best of my understanding, the most important point of the paper, the data supporting it should be presented and properly analyzed!

4) Pvd Super producers: the authors defined a subset of strains which produce more pvd than their ancestors (based on longitudinal sampling and genomic sequences). Super-production occurs in 50% of lineages and rather early (within ~2 years) in 33% of lineages. In a minority of super-producer lineages, the authors suggest that it is due to mutations in general regulators (e.g., lasR, mucA, gac). The authors suggest that this is an adaptation due to increased requirements for iron in the lungs. Finally, the author observed that a large fraction of super-producers became null producers (8/24) compared to strains that do not super-produce (4/21). We have several remarks on this part:

a) If super-production results from mutations in general regulators such as lasR or mucA, then pvd over-production might be a pleiotropic side effect of selection for a different trait dependent on the global regulator. This option is not mentioned in the text.

b) Super-producers by themselves are also super-cooperators which will be cheated by their wild-type pvd producer ancestors in their social environment. Something needs to explain the increase of cooperation. Selection for private pleiotropic effects of the mutation (section a) might explain this increased cooperation. Again, social evolution is not discussed in this context.

c) The observation that pvd- mutant phenotype arise more often in super-producers (8/24) than in wild-type producers (4/21) seems to be rather marginal to me. I'm not sure what is the statistical significance analysis, but if one simply assumes binomial distribution around the mean (12/45=26%), then with 21 samples, the chance of obtaining 4 or less events is ~30% and the chances of obtaining 8 events or more in 24 samples is ~40%. Therefore, I am not convinced that this difference is not occurring simply by chance.

5) Sensitivity to heme levels. If I understand correctly, the authors show in Figure 5 the average final OD for multiple related pairs which encode for the phu+ or phu-wt alleles (all in pvd- background) for different levels of heme (1, 2, 5, 10μM). I thought I got the picture clearly, but looking at the supplementary information and figure got me completely confused. The authors refer to six pairs of strains, for which they measured the growth in 2 and 5 μm of heme, but it is unclear on what was 1 and 10 μm measured. Also, while Figure 5 indicates that the phu-wt allele has a growth advantage in 5uM, Appendix 1—figure 7 shows that all the statistically significant (or marginally significant) effects in specific pairs, go on the other direction – phu+ grow better than phu-wt. I couldn't figure out how to resolve these two values in a satisfying manner.

[Editors' note: further revisions were requested prior to acceptance, as described below.]

Thank you for resubmitting your work entitled "Privatisation rescues essential function following loss of cooperation" for further consideration at *eLife*. Your revised article has been favorably evaluated by Ian Baldwin (Senior Editor), a Reviewing Editor, and one reviewer.

The manuscript has been improved but there are some remaining issues that need to be addressed before acceptance, as outlined below:

*Reviewer #2:*

In this version, Andersen et al. tried to address the main criticism brought forward in the previous version. All in all, I find the data and figures added to the work to satisfy my main points of criticism. I still have few comments here and there, where I think analysis could be improved or representation could be simplified for the reader.

I would go over my previous main reservations and whether I am satisfied from the corrections done by the authors:

1) Connection between pvd receptor mutations and phu+ phenotypic change. The authors now added the analysis of pvd reception mutations and show a good correspondence between the phu+ promoter mutations and accumulation of pvd reception mutations. I think this supports the notion that cheating on pvd is not occurring upon the phu+ phenotypic change.

2) Social interactions. This is the main point of the paper – loss of pyoverdine production from the social environment and not only from a specific lineage explains the rise of phu+ phenotype. The authors now provide Appendix 1—figure 3A, B to directly show the data. I think this is a good start, but I would suggest that the authors would provide a more patient-centric view of the data and not lineage-centric. The problem with the current representation is that different lineages in the same patient are scattered throughout the two figures, complicating comparison. To look at the data of a specific patient, I had to scan the two figures to find all lineages sampled in this patient and then mentally order them by the sample time, to try and see whether the claim (pvd loss from all samples prior to phu+ mutation) can be verified. I suggest that in addition to the current representation, the authors should show a representation were the data samples for a single patient are clustered together with ascending years and the phenotypic/genetic status of each sample is clearly given (using the same mutation annotation as in the current figure). This would facilitate direct visual estimation of the social data in a given lineage. I don't think there is a need to show patients which have been sampled once with a single clone, as nothing much can be said about their social environment (many such patients are shown in Appendix 1—figure 2).

3) Super producers: the statistical claim of the author is that pvd cheaters are not only preferentially arising in the super-producers, but that they also arise more quickly (i.e., that the rate of appearance of the mutations is higher. This is clear from Figure 1 but is not mentioned appropriately in the text (subsection “More cooperative populations are more vulnerable to exploitation”, third paragraph). I think the authors should also mention that the mean time for appearance of a pvd- mutation was quicker for the eight mutations that arose in the presence of super producer than the mean time for the four that arose without it.

4) Number of generations. The reviewers have cut their estimates considerably, but still rely on an estimation of doubling time of ~70 minutes, which I still find unconvincing – it will either imply that the whole mocus volume of the lungs is replaced every 70 minutes (as in a chemostat, where growth is determined by clearance rate), or that there is a massive level of cell death alongside fast cell growth (and then the chemostat assumptions fail, as average growth does not reflect cell divisions). I am no expert in CF, but both things seem unlikely, though I might be completely wrong on that. In any case, I don't find this to be very important. I would just throw away this estimate, but if the authors choose to keep it, I would not object.

---

## [Author Response]

Summary:This manuscript by Andersen et al. describes the evolution of Pseudomonas aeruginosa in the lungs of people with CF with a particular focus on evolutionary contingency around cooperative iron acquisition systems. The authors analyze isolates obtained from CF patients sputum in two cohorts of patients with longitudinal sampling and conclude that a series of invasions caused by social evolution take place in the CF lung: first, social cheats invade in siderophore (public good) producing populations, leading to the demise of public good producers. Surprisingly, this event is followed by the raise in frequency of private iron acquisition strategies, which should in principle be able to compete against cheaters. This type evolutionary dynamics, reminiscent of a rock-paper-scissors game, are fundamentally interesting and potentially relevant in the context of pathogenesis, as they could shed light on the selective pressures that impact the evolution of Pseudomonas aeruginosa in CF patients.

Point of clarification: this mis-represents our findings. Cells adopting private iron uptake strategies may also be pyoverdine cheaters, in fact, we find that they always are. There are not three distinct strategies as in the rock-paper-scissors game and this is not a good model for our observations. Instead, the population “escapes” coop cheat dynamics completely by switching to a private mechanism. To our knowledge, we are the first to report this strategy as a response to cheat invasion.

[…] Essential revisions:1) Clarity of the data: It is very hard to follow the numbers in this work! Numbers are differently represented between figures and text and often do not seem to match. For example, In Figure 1 and the relevant text they claim they observed 14 pvd- cheating events (9+5), while later on they claim they had 23 events (14+9) (subsection “Evolution of iron acquisition systems in the lung – the sequence of events”). What is the relation between these two numbers? The authors need to find a better way to represent the data in a clearer fashion. We strongly suggest to have a new figure with a visualization of the number of strains per time point, per patient and their phenotypes/genotypes.

The study includes two collections of isolates. One from young patients infected primarily with environmental clone types, sampled extensively within patients for max 10 years, and one from patients infected by two transmissible clone types, sampled extensively between patients for >30 years. In the latter there are few samples from individual patients, and longer time between samples. In the analysis of upregulation of pyoverdine we only used samples from the frequently sampled young patients in order to track the pyoverdine dynamics within patients as closely as possible. This information is in the Materials and methods, but to clarify we have now mentioned the two isolate collections in the Introduction (last paragraph), provided more detail in the Materials and methods (subsection “Isolate collections”, first paragraph; subsection “Changes in pyoverdine production during infection”, first paragraph; subsection “Transitions between public and private iron uptake”, first paragraph), and further described in the Results and Discussion which isolates are used when (subsection “More cooperative populations are more vulnerable to exploitation”, first paragraph; subsection “Evolution of iron acquisition systems in the lung – the sequence of events”, first paragraph; subsection “Cheat invasion is associated with a subsequent switch to private iron uptake mechanism”, first paragraph).

2) Relation to previous work by the same group: The main problem I see here is the relation between this work and the mBio paper by Marvig et al., 2014, and the PNAS paper by Andersen et al., 2015, both done by the same research groups involved in this work and the latter with the same first author as the current work. The authors often refer to these two papers in the current manuscript, but reading these two papers reveal that most of the basic observations mentioned in the current text were already made in these two papers. It is therefore unclear what is the novelty of this manuscript.

We have reworded the Introduction (last paragraph), and the Discussion (subsection “Conclusions”) to clarify the novel contribution made by the current submission. This is the first study that we know of to reveal privatisation as a strategy to survive the loss of an essential trait through cheat invasion. Neither of the papers cited address this question, neither is it possible to address this question with the data they present. No other study has shown this in bacteria or in any other system. Bacterial infections offer unique opportunities to observe long-term evolutionary dynamics that are not possible in other taxonomic groups that are studied by the research community interested in social behaviours, e.g. the social insects, vertebrates. In no other system has it been possible to observe the invasion of cheats and the following evolutionary response in a natural population. We expect our results to be of general interest beyond the microbiology community.

Marvig et al., 2014, showed that intergenic mutations in the phu system are common, found in isolates that do not produce pyoverdine, and cause an increased growth rate in the presence of heme as an iron source. The results are interpreted as host adaptation to increased availability of heme as an iron source. In Andersen et al., 2015, we reported the first evidence that cheating is responsible for long-term loss of pyoverdine in a population of *P.*

*aeruginosa* infecting the CF lung. Importantly, we were able to present data that helped to reject the alternative, widely-held hypothesis: that pyoverdine loss was due to redundancy as iron levels changed through infection. We provided the first evidence of cheat invasion in a natural population of infectious bacteria.

The current paper reports the following novel findings: (1) increased cooperation facilitates the invasion of cheats, and (2) bacterial populations respond to the loss of an essential trait through cheating by switching to a private mechanism for iron uptake.

a) Cheating on pyoverdine production: Andersen et al., 2015, have shown that pvd mutations first arise in pvd production pathway (mainly pvdS) and later in the reception pathway, only once the pvd- production mutants are fixed in the population (based on other strains obtained in the same sample and in previous samples of the same patient). They hypothesize that production mutants are cheaters and that pvd reception mutants are favorable due to the cost of reception in the absence of pvd-related benefit. Several points should be made with respect to this manuscript.i) First, it seems that the basic observation of cheating on pvd production has already been made. The only thing that is added in this manuscript is the observation regarding the relation between cheating and super-producers, to which I have several reservations (see separate point below).

The observation of cheating is not claimed as novel in this manuscript but stated as a finding of our previous article (Introduction, fourth paragraph).

ii) In the current work, only production mutants are discussed, while reception mutants are not mentioned at all. Since both pvd reception mutants and phu+ mutants are correlated with fixation of the pvd production mutants, it is surprising that reception mutants were not discussed and compared with phu+ mutants.

The Phu+ mutants are found only when pyoverdine production has been lost from the population, i.e. when cheating is no longer an option. We have elaborated: “Knock-out of the pyoverdine receptor is, however, not expected to be a selective force in *phu* upregulation. Rather, the data suggest that both occur because pyoverdine is no longer available in the environment, selecting for privatisation of iron uptake and elimination of the costly receptor”. The presence or absence of pyoverdine receptor mutations have been added to Appendix 1—figure 3.

b) Marvig et al., has already identified the phu+ promoter mutation and the fact that they arise only in pvd production mutants, though their work was done on a smaller number of lineages. Therefore, it is not clear what is the added value of the current work? Is it the demonstration that phu+ alleles are only beneficial at low heme levels (see comments on the experiments in a separate point)? Is the claim for novelty in this work depends on the enlarged statistical power? Or is it the claim that phu+ promoter mutants arise only when pvd+ phenotype is completely eliminated from the social environment and not only from the specific lineage? See comments on sociality below for the last option. Altogether, the distinction between the current work and previous works of the group are not sufficiently clear.

See comment above. The current manuscript is clearly distinct from Marvig et al. in the hypotheses tested, which concern selective forces responsible for the observed changes, i.e. social interactions and not solely host adaptation. It is significantly less challenging to report the presence of a mutation than it is to present evidence for why a mutation has arisen, and why it has spread. This is what we do here.

3) Social interaction: It is hard to understand when do the authors discuss social environment within the given lineage (i.e., when discussing interaction between a mutant and its ancestor) and when do they discuss social environment based on multiple strains sampled from the same patient at the same time.

The social environment is characterized as the co-occurring genotypes within a clone type, i.e. mutants and ancestor, as described (“the social environment (the *P. aeruginosa* clonal lineage infecting a host)”) and now as further clarified in the first paragraph of the subsection “Isolate collections”. We focus on within clone-type events as continuous overlap between clone types within patients occurred rarely. The distinction has been further highlighted in the first paragraph of the subsection “Changes in pyoverdine production during infection” and Appendix 1—figures 1 and 2.

While the authors claim social explanations for their observation, much of the analysis in the text is not based on direct social observations, but on comparisons between a mutant and its ancestors (e.g., Figure 1, 2, 3). The two places where a social context is mentioned are:a) When they discuss the social environment of pvd- cheats, the authors mention that 14 events are not fixed while 9 other are fixed (subsection “Evolution of iron acquisition systems in the lung – the sequence of events”).b) Later, the authors mention that the phu+ mutations occurred significantly more when Pvd was lost, but here it is unclear whether they mean that Pvd was lost from the genome of the strains, or from the total social environment (subsection “Cheat invasion is associated with a subsequent switch to private iron uptake mechanism”), as they refer to Figure 2 which, to the best of my understanding, discussed only changes within a given genome and not the social context. On the other hand, they specifically mention that in all but three cases (I guess therefore that in 31-3=28 cases, but again – distinction between lineages and patients is unclear), phu+ arise when no pvd+ strains were observed at all.In both cases there is insufficient information given on the social environment – how many strains were analyzed in any given sample. This data in neither presented nor analyzed in anyway whatsoever in the text. Now, this is obviously important – I would find the claim for fixation of a given genotype (or the elimination of another) to be much stronger if 10 isolates were sampled from the same patient at the same time than if only two samples were taken (or one!). Yet, the social data is completely absent, so I do not know how to interpret the claim for fixation of the cheater. As this is, to the best of my understanding, the most important point of the paper, the data supporting it should be presented and properly analyzed!

The number of samples and extent of sampling from each patient is available in Supplementary file 2. To further clarify the sampling regimen we have included Appendix 1—figures 1 and 2 showing sampling time for the clone types that were longitudinally sampled from patients, highlighting over-producers and non-producers from the children’s collection (Appendix 1—figure 1; Clone types that were only sampled once are not included in the analysis or the figure, l. 183-184). For the transmissible clone types both the sampling time across patients and phylogenies with selected pheno – and genotypes are shown (Appendix 1—figures 2 and 3).

During patient sampling sputum was collected and cultured, and 1-8 colonies were selected and stored for further analyses, as clarified in the first paragraph of the subsection “Isolate collections”. Therefore, the sequencing effort was partly determined by how the samples were originally collected, and partly by prioritizing extent of sampling, rather than depth, for the children’s collection. As mentioned in the introduction, a subsequent metagenomic study showed that the longitudinal samples represent the standing diversity well, and as we explain “The likelihood that we do not capture all co-infecting strains in our sample, leading to mis-classification of isolates as belonging to an environment without pyoverdine, will homogenise the sample groups and hence obscure any differences. Sampling errors of this kind will, therefore, hinder our ability to detect an effect (Type I error).” While more samples are of course always desirable, we feel very confident in the presented result, in particular because of the high degree of convergent evolution of the genotypes we focus on, across clone types and patients.

Throughout the text we have clarified that some clone types occur in multiple patients, while some patients are infected by multiple clone types (subsection “Isolate collections”, first paragraph; subsection “Changes in pyoverdine production during infection”, last paragraph; subsection “More cooperative populations are more vulnerable to exploitation”, second paragraph; Supplementary file 2), which may have caused confusion in the presentation of results on clonal lineages in patients.

We define samples as co-occurring if they are sampled < 1 year apart (subsection “Evolution of iron acquisition systems in the lung – the sequence of events”, last paragraph). That means that if a non-producer is sampled and a producer sampled < 1 year prior, or subsequently, then the social environment of that non-producer is considered to be cooperative. When no more producers are sampled, cheating is considered to have gone to fixation.

4) Pvd Super producers: the authors defined a subset of strains which produce more pvd than their ancestors (based on longitudinal sampling and genomic sequences). Super-production occurs in 50% of lineages and rather early (within ~2 years) in 33% of lineages. In a minority of super-producer lineages, the authors suggest that it is due to mutations in general regulators (e.g., lasR, mucA, gac). The authors suggest that this is an adaptation due to increased requirements for iron in the lungs. Finally, the author observed that a large fraction of super-producers became null producers (8/24) compared to strains that do not super-produce (4/21). We have several remarks on this part:

We do not state that super-producers become cheats, rather that cheats are more likely to be sampled following the observation of super-cooperators (subsection 2 More cooperative populations are more vulnerable to exploitation”, third paragraph). As the genetic basis for increased production is not identified, we do not know if a sampled cheat evolved from a super-cooperator or from an isolate with WT pyoverdine production. We have clarified this in the third paragraph of the subsection “More cooperative populations are more vulnerable to exploitation”.

a) If super-production results from mutations in general regulators such as lasR or mucA, then pvd over-production might be a pleiotropic side effect of selection for a different trait dependent on the global regulator. This option is not mentioned in the text.

We agree that over-production of pyoverdine may be a part of an overall “acute infection” phenotype. We have further highlighted in the text that this may be as a pleiotropic effect or beneficial trait (subsection “More cooperative populations are more vulnerable to exploitation”, second paragraph). Regardless of whether the selective force is iron uptake or host colonization, the effects of upregulation on social interaction may be the same.

b) Super-producers by themselves are also super-cooperators which will be cheated by their wild-type pvd producer ancestors in their social environment. Something needs to explain the increase of cooperation. Selection for private pleiotropic effects of the mutation (section a) might explain this increased cooperation. Again, social evolution is not discussed in this context.

We present three hypotheses for increased pyoverdine production, i) iron limitation, ii) inter-species iron competition, and iii) selection on an acute infection phenotype (subsection “More cooperative populations are more vulnerable to exploitation”, second paragraph). Organisms with WT level production of pyoverdine may also exploit super-cooperators but the benefit of exploitation increases with an increased difference in pyoverdine production. Hence, cheats will have a greater benefit exploiting super-cooperators than WT, and than WT exploiting super-cooperators. We have clarified this in the last paragraph of the aforementioned subsection. As such, this paragraph is all about social evolution.

c) The observation that pvd- mutant phenotype arise more often in super-producers (8/24) than in wild-type producers (4/21) seems to be rather marginal to me. I'm not sure what is the statistical significance analysis, but if one simply assumes binomial distribution around the mean (12/45=26%), then with 21 samples, the chance of obtaining 4 or less events is ~30% and the chances of obtaining 8 events or more in 24 samples is ~40%. Therefore, I am not convinced that this difference is not occurring simply by chance.

The results of the statistical analysis are presented in the text (subsection “More cooperative populations are more vulnerable to exploitation”, third paragraph) and graphically in the inverse Kaplan-Meier plot in Figure 1. We have used a Survival analysis to take into account the timing of events and not simply correlate presence / absence. This has been further clarified in the Materials and methods and Results and Discussion (subsection “Changes in pyoverdine production during infection”, last paragraph; subsection “Cheat invasion is associated with a subsequent switch to private iron uptake mechanism”, last paragraph).

5) Sensitivity to heme levels. If I understand correctly, the authors show in Figure 5 the average final OD for multiple related pairs which encode for the phu+ or phu-wt alleles (all in pvd- background) for different levels of heme (1, 2, 5, 10μM). I thought I got the picture clearly, but looking at the supplementary information and figure got me completely confused. The authors refer to six pairs of strains, for which they measured the growth in 2 and 5 μm of heme, but it is unclear on what was 1 and 10μm measured. Also, while Figure 5 indicates that the phu-wt allele has a growth advantage in 5uM, Appendix 1—figure 7 shows that all the statistically significant (or marginally significant) effects in specific pairs, go on the other direction – phu+ grow better than phu-wt. I couldn't figure out how to resolve these two values in a satisfying manner.

As stated in the text Figure 5 shows growth measurements of an isogenic pair of clinical isolates, differing only in the presence of a phuS//phuR mutation (subsection “Uptake of heme following phuS//phuR mutations”, first paragraph; subsection “Why is private heme uptake only upregulated if cooperation is lost?”, first paragraph). Here we see a clear difference in growth dependent on the heme availability. In Appendix 1—figure 4 we show measurements from pairs of clinical isolates that differ in the presence of phuS//phuR mutations (subsection “Uptake of heme following phuS//phuR mutations”, last paragraph). Although the isolates in a pair are the closest related we have, they have been sampled many years apart and differ by many additional mutations that likely affect growth (Appendix). These isolates overall show higher growth at 5 μm heme added, compared to at 2.5 μm for the isogenic pair. We have added text to highlight this difference (subsection “Why is private heme uptake only upregulated if cooperation is lost?”, last paragraph; Appendix). In the Methods we have switched the order of presenting the isogenic pair and the clinical isolates for clarity.

[Editors' note: further revisions were requested prior to acceptance, as described below.]

The manuscript has been improved but there are some remaining issues that need to be addressed before acceptance, as outlined below:Reviewer #2:In this version, Andersen et al. tried to address the main criticism brought forward in the previous version. All in all, I find the data and figures added to the work to satisfy my main points of criticism. I still have few comments here and there, where I think analysis could be improved or representation could be simplified for the reader.I would go over my previous main reservations and whether I am satisfied from the corrections done by the authors:1) Connection between pvd receptor mutations and phu+ phenotypic change. The authors now added the analysis of pvd reception mutations and show a good correspondence between the phu+ promoter mutations and accumulation of pvd reception mutations. I think this supports the notion that cheating on pvd is not occurring upon the phu+ phenotypic change.

Yes, thank you for the suggestion. We agree that this has helped to clarify our results.

2) Social interactions. This is the main point of the paper – loss of pyoverdine production from the social environment and not only from a specific lineage explains the rise of phu+ phenotype. The authors now provide Appendix 1—figure 3A, B to directly show the data. I think this is a good start, but I would suggest that the authors would provide a more patient-centric view of the data and not lineage-centric. The problem with the current representation is that different lineages in the same patient are scattered throughout the two figures, complicating comparison. To look at the data of a specific patient, I had to scan the two figures to find all lineages sampled in this patient and then mentally order them by the sample time, to try and see whether the claim (pvd loss from all samples prior to phu+ mutation) can be verified. I suggest that in addition to the current representation, the authors should show a representation were the data samples for a single patient are clustered together with ascending years and the phenotypic/genetic status of each sample is clearly given (using the same mutation annotation as in the current figure). This would facilitate direct visual estimation of the social data in a given lineage. I don't think there is a need to show patients which have been sampled once with a single clone, as nothing much can be said about their social environment (many such patients are shown in Appendix 1—figure 2).

We agree with the reviewer that this is a useful way to present the data and have provided a patient-centric overview in Appendix 1—figure 4, showing the phenotypes and sampling time for the transmissible clone types DK1 and DK2. The two additional clone types that acquire phu mutations have also been included. We discuss the within-patient social environment in the Appendix text.

3) Super producers: the statistical claim of the author is that pvd cheaters are not only preferentially arising in the super-producers, but that they also arise more quickly (i.e., that the rate of appearance of the mutations is higher. This is clear from Figure 1 but is not mentioned appropriately in the text (subsection “More cooperative populations are more vulnerable to exploitation”, third paragraph). I think the authors should also mention that the mean time for appearance of a pvd- mutation was quicker for the eight mutations that arose in the presence of super producer than the mean time for the four that arose without it.

To clarify, the cleanest prediction that we can make is that cheats are more likely to arise in the presence of super-cooperators, compared to when they are absent. This is the prediction we test with our “survival analysis”. Predicting a difference in time before a mutation arises is more tenuous: what to compare exactly? Time since super-producer first arrives and appearance of a cheat, or time since first colonization and the appearance of a cheat in both cases? As we are interested in the potential effect of super-cooperators in the analysis, we compare the time since first sampling of super-cooperator and appearance of a cheat, to time from colonisation to cheat appearance in the absence of super-cooperators. The reviewer highlights that the difference in time axes on the figure can lead to the mistaken impression that cheats arise quicker in the presence of super-producers. In order to avoid misleading readers, we now explicitly state in the text that there is no significant difference in time to first appearance (subsection “More cooperative populations are more vulnerable to exploitation”, third paragraph), and in the Materials and methods we further clarify “If an over-producer was present, the length of time from that sampling to a non-producer was observed […]”. Thank you to the reviewer for pointing out that this would be helpful.

4) Number of generations. The reviewers have cut their estimates considerably, but still rely on an estimation of doubling time of ~70 minutes, which I still find unconvincing – it will either imply that the whole mocus volume of the lungs is replaced every 70 minutes (as in a chemostat, where growth is determined by clearance rate), or that there is a massive level of cell death alongside fast cell growth (and then the chemostat assumptions fail, as average growth does not reflect cell divisions). I am no expert in CF, but both things seem unlikely, though I might be completely wrong on that. In any case, I don't find this to be very important. I would just throw away this estimate, but if the authors choose to keep it, I would not object.

Our estimate for generation times is based on the experimental work by Yang et al., 2011 on the very isolate collection that we work on. It is, to our knowledge, the best available for the CF environment. This is highlighted in the legend (Supplementary file 1A legend).